# Reovirus directly engages integrin to recruit clathrin for entry into host cells

Melanie Koehler [1,6], Simon J. L. Petitjean[1,6], Jinsung Yang [1,6], Pavithra Aravamudhan[2,3], Xayathed Somoulay[2,3], Cristina Lo Giudice [1], Mégane A. Poncin[1], Andra C. Dumitru[1], Terence S. Dermody[2,3,4 ✉] & David Alsteens [1,5 ✉]

Reovirus infection requires the concerted action of viral and host factors to promote cell entry. After interaction of reovirus attachment protein σ1 with cell-surface carbohydrates and proteinaceous receptors, additional host factors mediate virus internalization. In particular, β1 integrin is required for endocytosis of reovirus virions following junctional adhesion molecule A (JAM-A) binding. While integrin-binding motifs in the surface-exposed region of reovirus capsid protein λ2 are thought to mediate integrin interaction, evidence for direct β1 integrin-reovirus interactions and knowledge of how integrins function to mediate reovirus entry is lacking. Here, we use single-virus force spectroscopy and confocal microscopy to discover a direct interaction between reovirus and β1 integrins. Comparison of interactions between reovirus disassembly intermediates as well as mutants and β1 integrin show that λ2 is the integrin ligand. Finally, using fluidic force microscopy, we demonstrate a functional role for β1 integrin interaction in promoting clathrin recruitment to cell-bound reovirus. Our study demonstrates a direct interaction between reovirus and β1 integrins and offers insights into the mechanism of reovirus cell entry. These results provide new perspectives for the development of efficacious antiviral therapeutics and the engineering of improved viral gene delivery and oncolytic vectors.

[1] Louvain Institute of Biomolecular Science and Technology, Université catholique de Louvain, Louvain-la-Neuve, Belgium. [2] Department of Pediatrics, University of Pittsburgh School of Medicine, Pittsburgh, PA, USA. [3] Institute of Infection, Inflammation and Immunity, UPMC Children's Hospital of Pittsburgh, Pittsburgh, PA, USA. [4] Department of Microbiology and Molecular Genetics, University of Pittsburgh School of Medicine, Pittsburgh, PA, USA. [5] Walloon Excellence in Life sciences and Biotechnology (WELBIO), Wavre, Belgium. [6] These authors contributed equally: Melanie Koehler, Simon J. L. Petitjean, Jinsung Yang. ✉email: terence.dermody@chp.edu; david.alsteens@uclouvain.be

As obligate intracellular microbes, viruses must cross the plasma membrane to enter host cells and initiate infection. Viruses use various strategies and exploit physiological cellular processes for entry. Infection is initiated by contact of the virus with the cell surface, followed by attachment to specific cell-surface molecules. Nonspecific and low-affinity interactions with abundant cell-surface molecules, which are sometimes called attachment factors, concentrate the virus on the plasma membrane to facilitate subsequent high-affinity interactions with specific receptors that trigger virus internalization[1,2]. While attachment steps (e.g., to cell-surface glycans) are mostly shared among viruses, internalization pathways are more diverse and depend on the virus and receptors involved. In particular, internalization of nonenveloped viruses is more complex than for enveloped viruses, as they lack a membrane to allow fusion of viral and cellular surfaces for delivery of the viral payload into the cytosol. Therefore, mechanisms underlying cell entry for many nonenveloped viruses remain poorly understood.

Mammalian orthoreoviruses (reoviruses) are nonenveloped, double-stranded RNA viruses that infect a broad range of mammalian hosts in nature. Reovirus infection, although mostly asymptomatic in humans, has been linked to the loss of oral tolerance to gluten implicated in celiac disease[3]. Conversely, the oncolytic properties[4] of reovirus make them potential tools for cancer therapy, and candidate strains have shown promising results in clinical trials against refractory human cancers[5]. Reovirus virions contain two concentric protein shells, an outer capsid and inner core[6]. Outer-capsid proteins, μ1, σ1, and σ3, as well as the core protein λ2, function in reovirus cell entry (Fig. 1a). Protein λ2 forms pentameric turrets that provide a base for insertion of the filamentous attachment protein, σ1. Virions undergo progressive proteolytic disassembly during cell entry forming infectious subvirion particles (ISVPs) (Fig. 1c) and cores (Fig. 1d)[7], that vary from intact virions in protein structure and composition. Both ISVPs and cores can be produced in vitro by limited proteolytic digestion of reovirus virions. ISVPs lack the major outer-capsid protein σ3 and contain a cleaved form of μ1 as well as an extended conformer of σ1[8]. Further removal of μ1 and σ1 from ISVPs generates viral cores.

The initial attachment of reovirus virions to cell-surface receptors is mediated by σ1, which has binding sites for α-linked sialylated glycans (α-SA)[9] and junctional adhesion molecule-A (JAM-A)[10]. The two commonly studied reovirus serotypes, T1 and T3, engage glycans and JAM-A, which act in concert through an adhesion-strengthening mechanism. Initial low-affinity interaction of σ1 with α-SA enhances JAM-A binding capacity, possibly by triggering conformational changes in σ1 (Fig. 1b)[11]. However, the cytoplasmic tail of JAM-A is dispensable for reovirus internalization[12]. Instead, β1 integrin mediates

reovirus internalization, likely through clathrin-dependent endocytosis. NPXY motifs in the cytoplasmic tail of β1 integrin, which function in recruitment of adaptor proteins and clathrin for endocytosis and serve as sorting signal for internalized cargo, are required for efficient reovirus sorting in the endocytic compartment and infection of cells[13]. Several other pathogenic microbes, including adenovirus, which uses similar attachment mechanisms as reovirus[14], usurp the adhesion and signaling properties of integrins to bind and enter host cells[15–17]. In general, integrins function as an α/β subunit pair, and metal ions such as $Mn^{2+}$, $Mg^{2+}$, and $Ca^{2+}$ promote different conformations and ligand binding affinities[18]. In addition, ligand binding to integrins activates signal transduction and promotes internalization through multiple mechanisms including clathrin-mediated endocytosis[19].

From a structural point of view, the reovirus λ2 protein has conserved integrin-binding motifs (IBMs) RGD and KGE that are solvent exposed in the virion[20–22]. However, there is no evidence for direct interactions between λ2 and integrin. Interestingly, although ISVPs also contain λ2, these particles do not require integrin for cell entry and are thought to directly penetrate the plasma membrane or use a caveolin-dependent uptake mechanism[23]. While remarkable progress has been made in deciphering mechanisms underlying reovirus cell attachment, the function of integrins in viral entry remains elusive.

Here, we used force-distance (FD) curve-based atomic-force microscopy (AFM) to investigate reovirus binding to β1 integrin. We demonstrate direct and specific interactions between reovirus and β1 integrin immobilized on model surfaces and expressed in living cells. We also show that the dynamics of virus-integrin interactions are influenced by divalent cations ($Mn^{2+}$, $Mg^{2+}$, $Ca^{2+}$) that modulate the conformation and affinity of β1 integrins for ligands[18]. By further comparing β1 integrin interactions with either reovirus virions, ISVPs, or cores, we provide new insights into the capacity of each particle form to bind β1 integrin and demonstrate a function for λ2, more specifically the λ2 RGD and KGE motifs, as the viral ligand for integrin. Finally, by using advanced imaging approaches, we discovered that integrin engagement by reovirus leads to dynamic recruitment of clathrin to the plasma membrane which may in turn facilitate endocytosis. Taken together, these results demonstrate that direct interaction of reovirus with β1 integrin mediates clathrin recruitment on host cells.

## Results

**Reovirus can directly engage β1 integrin.** To investigate whether reovirus can bind integrins, we used FD-curve-based AFM[11,24] to quantify the binding strength of reovirus to β1 integrins both on

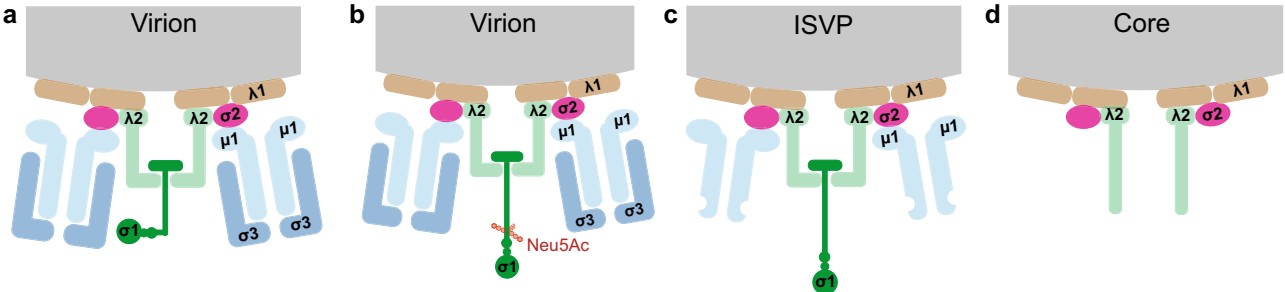

**Fig. 1 Schematic of capsid composition of reovirus particles. a–d** Diagrams of cross-sections of reovirus virions, infectious subvirion particles (ISVPs), and cores. Schematics depict the arrangement and conformation of structural proteins in the double-layered shells of the virions in the absence (**a**) and presence (**b**) of Neu5Ac (triggering a conformational change of σ1); removal of σ3, cleavage of μ1, and rearrangement of σ1 in ISVPs (**c**); and removal of μ1 fragments, loss of σ1, and opening of the λ2 turret in cores (**d**).

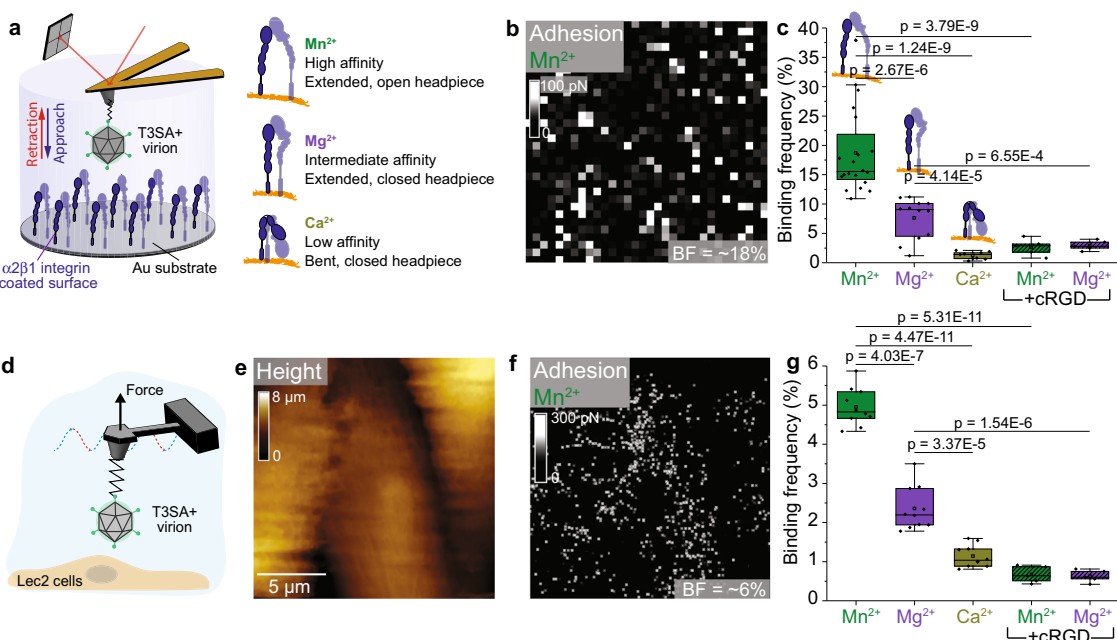

**Fig. 2 Reovirus directly engages β1 integrins in a cation-dependent manner.** Studies were conducted using model surfaces (**a–c**) and living cells (**d–g**). **a** Experimental set up (left) to probe cation-dependent (right) virion-α2β1 integrin interactions using a model surface. **b** The adhesion map shows interaction forces (white pixels) between virions and integrins in the presence of $Mn^{2+}$, which induces an extended integrin conformation. **c** Box plot of specific binding frequencies (BF) measured using AFM between virions and integrins in the presence of divalent cations and following injection of cRGD. **d** FD-based AFM setup to study T3SA+ binding to Lec2 cells, which express β1 integrins. Height (**e**) and corresponding adhesion maps (**f**) of the imaged area show specific binding of reovirus T3SA+ to living Lec2 cells in the presence of $Mn^{2+}$ (white pixels). For better visibility, the pixel size in the adhesion image was enlarged two-fold. AFM images were acquired using an oscillation frequency of 0.25 kHz and amplitude of 750 nm under cell culture conditions. **g** Box plot shows BF of T3SA+ virions to β1 integrins on living Lec2 cells under conditions shown. The horizontal line within the box indicates the median, boundaries of the box indicate the 25th and 75th percentile, and whiskers indicate the highest and lowest values. Open square within each box indicates the mean. All data are representative of $N = 5$ independent experiments. *P* values were determined by two-sample *t*-test using Origin.

model surfaces and living cells. To mimic cell-surface β1 integrins in vitro, we covalently immobilized α2β1 integrins on gold-coated surfaces via NHS/EDC chemistry. These model surfaces were imaged using AFM, and the thickness of the grafted integrin layer was measured by a scratching experiment, which revealed a deposited layer of ~3.0 ± 0.5 nm (see methods and Supplementary Fig. 1a, cross-section in the inset). Reovirus interaction with integrin was probed by covalently attaching purified reovirus strain T3SA+ virions (α-SA binding strain) to the free end of a long, polyethylene glycol (PEG)$_{27}$ linker chemically coupled to an AFM tip[24,25] (Supplementary Fig. 1b, shows a single virion at the tip apex). To investigate how integrin conformation influences ligand binding, FD curves were recorded in the presence of divalent cations to assess binding capacity and strength of reovirus binding to integrin (Fig. 2a–c). In the presence of $Mn^{2+}$, which promotes an integrin conformation with high ligand affinity, specific adhesion events were observed in 19 ± 3% (mean ± standard deviation [S.D.], $N = 20480$) of retraction FD curves for specific events (i.e., with bond rupture distances >5 nm, which corresponds to the extension of the PEG linker) (Fig. 2c, green box). In the presence of $Mg^{2+}$, which stabilizes β1 integrin in its intermediate-affinity conformation, we observed a significant decrease in the binding frequency (BF) to 8 ± 3% (mean ± S.D., $N = 12288$) (Fig. 2c, purple box). A further decrease in BF to 1 ± 1% (mean ± S.D., $N = 9216$) was observed in the presence of $Ca^{2+}$ (Fig. 2c, dark yellow box), which promotes a low-affinity conformation of integrin. These results reveal a cation-dependent interaction between reovirus and integrins. To test the specificity of the observed interactions, we injected cyclic arginine-glycine-aspartate (cRGD), a cyclic peptide that binds β1 integrin with high-affinity, prior to reovirus. We anticipated that cRGD would

compete with β1 integrin for reovirus interaction. Injection of cRGD reduced the BF to <3% ($N = 9216$) in the presence of either $Mn^{2+}$ or $Mg^{2+}$ (Fig. 2c, dashed boxes). This result, coupled with the cation-dependency of binding, confirms that β1 integrin can directly interact with reovirus in a conformation-dependent manner.

Next, we probed reovirus-integrin interactions in a more physiologically relevant context using living Lec2 cells (Fig. 2d–g). Lec2 is an α-SA-deficient cell line derived from Chinese hamster ovary (CHO) cells that does not express SA[11], facilitating investigation of specific interactions with cell surface receptors. We first tested whether Lec2 cells express β1 integrin using immunofluorescence (Supplementary Fig. 2). β1 integrin expression was detected and mostly distributed to the bottom of the cell, which is consistent with its known function in mediating cell adhesion to the substrate. Reovirus interaction with β1 integrin was studied by recording AFM height images of Lec2 cells along with corresponding adhesion maps in the presence of the three different cations. Specific adhesion events localized to the cell surface were mapped (displayed as bright pixels), and the BF was extracted (Fig. 2e–g). Remarkably, Lec2 cells probed with T3SA+ showed the highest frequency of adhesion events (BF ~6%, $N = 10$ cells) in the presence of $Mn^{2+}$, followed by ~3% ($N = 10$ cells) in the presence of $Mg^{2+}$ (Supplementary Fig. 3e–h), and ~1% ($N = 10$ cells) in the presence of $Ca^{2+}$ (Supplementary Fig. 3i–l). These results confirm that the conformation of integrin on cells alters its binding affinity to reovirus. The addition of cRGD reduced the BF (~1%, $N = 5$ cells), thus confirming the specificity of the observed interactions (Supplementary Fig. 3m–p). Recording consecutive maps with the same AFM tip attached to T3SA+ validated that the virions

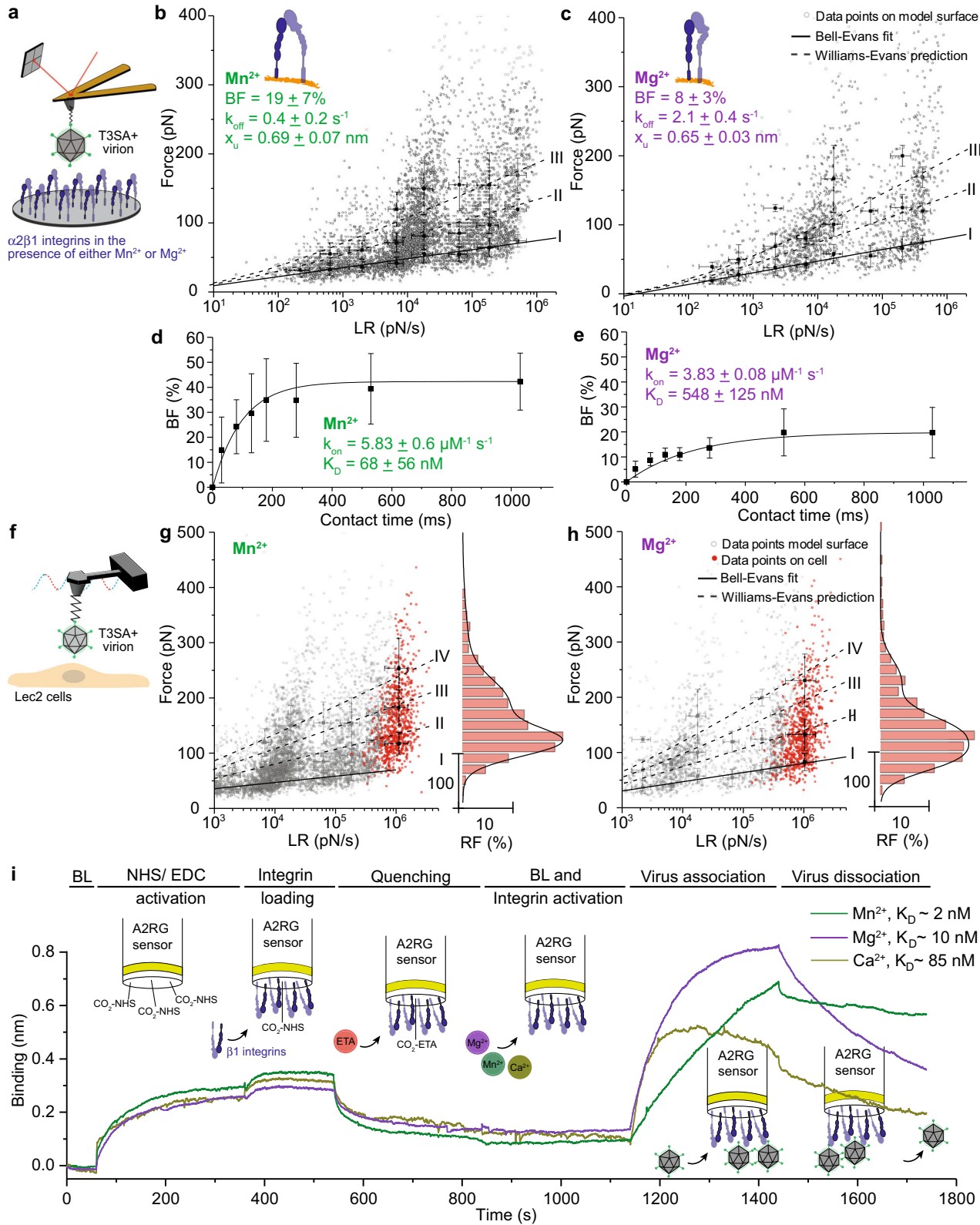

remained stable during scans (Supplementary Fig. 3a–d). The overall lower BF of reovirus compared with other integrin ligands[11] may be due to the preferential distribution of integrins to the basal cell surface, while only the apical surface is probed during AFM studies. Taken together, these data establish that reovirus specifically interacts with β1 integrin expressed on cells.

**Kinetic and thermodynamic insights into cation-dependent integrin-reovirus interactions**. To define the dynamics of reovirus-β1 integrin interaction, we extracted kinetic and thermodynamic parameters of the interaction using multiple approaches: AFM studies using model surfaces (Fig. 3b, c) and living cells (Fig. 3g, h) and biolayer interferometry (BLI) (Fig. 3i).

**Fig. 3 Kinetic and thermodynamic insights into cation-dependent reovirus-integrin interactions.** Studies were conducted using AFM on model surfaces (**a**–**e**) and living cells (**f**–**h**) and using BLI (**i**). **a** Schematic of experimental set up to probe cation-dependent reovirus interactions with α2β1 integrin-coated model surface. **b, c** Dynamic force spectroscopy (DFS) plots show distribution of average rupture forces determined at eight distinct loading rate (LR) ranges for interactions between T3SA+ virions and β1 integrin-coated model surface in the presence of $Mn^{2+}$ (for further details, see methods section) (**b**) or $Mg^{2+}$ (**c**). Data corresponding to single interactions were fit with the Bell–Evans (BE) model (I, black curve), providing average $k_{off}$ and $x_u$ values. Dashed lines represent predicted binding forces for two (II) and three (III) simultaneous uncorrelated interactions (Williams-Evans [WE] prediction). **d, e** The binding probability is plotted as a function of the contact time. Least-squares fit of the data to a mono-exponential decay curve (line) provides average kinetic on-rate ($k_{on}$) of the probed interaction. Comparison of $K_D$ ($k_{off}/k_{on}$) values shows that $Mn^{2+}$ increases the affinity of T3SA+ virions for β1 integrins. **f**–**h** Assessment of cation-dependent reovirus interaction with integrins expressed on living cells. DFS plots of data obtained using model surfaces (gray circles and living cells (red dots) in the presence of either $Mn^{2+}$ (**g**) or $Mg^{2+}$ (**h**). Histogram of the force distribution observed on cells and a multi-peak Gaussian fit of data ($n = 900$ data points) are shown at the side. Error bars indicate s.d. of the mean value. All data are representative of $N = 5$ independent experiments. **i** Sensorgrams obtained using biolayer interferometry (BLI) show interaction of T3SA+ with β1 integrins immobilized on amine-reactive biosensors under conditions shown.

**Table 1 Comparison of kinetic and thermodynamic parameters describing T3SA+ virion-β1 integrin interaction in the presence of either $Mn^{2+}$ or $Mg^{2+}$.**

| T3SA+ | $Mn^{2+}$ | $Mg^{2+}$ |
|---|---|---|
| BF [%] | 19 ± 7 | 8 ± 3 |
| $k_{off}$ [$s^{-1}$] | 0.4 ± 0.2 | 2.1 ± 0.4 |
| $k_{on}$ [$\mu M^{-1}\ s^{-1}$] | 5.83 ± 0.6 | 3.83 ± 0.08 |
| $K_D$ [nM] | 68 ± 56 | 548 ± 125 |
| $x_u$ [nm] | 0.69 ± 0.07 | 0.65 ± 0.03 |

Reovirus interaction with β1 integrin on model surfaces was probed using AFM at various loading rates (LRs) (i.e., force applied over time). Over the range of applied LRs, the β1 integrin-T3SA+ complex withstood forces in the range of 25–400 pN. Although forces in this range can affect conformational stability of proteins, reovirus particles remained mechanically stable, as evidenced by laser-scanning optical microscopy (Supplementary Fig. 1b) and control experiments showing thousands of stable interactions over several scans. Forces measured at different LRs were fitted to the Bell–Evans model[26], and the dissociation rate ($k_{off}$) and the distance to the transition state ($x_u$) of the interactions were extracted either in the presence of $Mn^{2+}$ (Fig. 3b) or $Mg^{2+}$ (Fig. 3c). A similar distance to the transition state was observed in the presence of both cations, indicating a similar binding geometry under these conditions. However, an increase in the dissociation rate was observed in the presence of $Mg^{2+}$ relative to $Mn^{2+}$ suggesting a less stable interaction in the presence of $Mg^{2+}$. In addition, multivalent interactions were also observed in the presence of both cations. Analysis of the data using the Williams-Evans (WE) prediction[27] (Fig. 3b, c, dashed curves II and III) shows that these multivalent interactions are uncorrelated bonds loaded in parallel. We hypothesize that these multivalent bonds are established between a single virion attached to the AFM tip, likely through λ2 molecules, and multiple β1 integrin molecules immobilized on the model surfaces for the following reasons: (i) λ2 is a pentamer with five possible binding sites for β1 integrin; (ii) each virion has 12 copies of the λ2 pentamer (corresponding to the 12 virion icosahedral vertices); (iii) the tip apex contains mostly single virions capable of interacting with the surface; (iv) virion unbinding occurs in a single step (a single rupture peak observed in the FD curves); and (v) the capacity to establish multivalent bonds correlates with the activation state of integrins. Finally, analysis of the influence of contact time on BF (Fig. 3d, e) shows a significantly higher ($P < 0.0001$) association rate ($k_{on}$) in the presence of $Mn^{2+}$ ($k_{on} = 5.83 ± 0.06\ \mu M^{-1}s^{-1}$) compared with $Mg^{2+}$ ($k_{on} = 3.83 ± 0.08\ \mu M^{-1}\ s^{-1}$), confirming that $Mn^{2+}$ facilitates β1 integrin binding by reovirus. Collectively, these experiments lead to the following

calculated equilibrium dissociation constants: $K_D = 68 ± 56$ nM in presence of $Mn^{2+}$ and $K_D = 548 ± 125$ nM in presence of $Mg^{2+}$ (Table 1). Interaction forces measured using living cells align with data gathered using model surfaces, further validating the observations (Fig. 3f–h).

To validate single-molecule measurements, we also used optical BLI and quantified T3SA+ virion binding to β1 integrins coated on amine-reactive biosensors. BLI data show that T3SA+ binds β1 integrins with high affinity ($K_D \sim 2$ nM) in the presence of $Mn^{2+}$ (Fig. 3i, green curve). As expected from single-molecule measurements, the presence of $Mg^{2+}$ (Fig. 3i, purple curve) or $Ca^{2+}$ (Fig. 3i, dark yellow curve) decreases reovirus avidity for β1 integrins with $K_D$ values of 10 nM and 85 nM, respectively. These data are consistent with results from the AFM experiments and show that the presence of $Mn^{2+}$ enhances the avidity of reovirus T3SA+ for β1 integrins. The overall higher (~30 times) dissociation constants observed using AFM compared with BLI may be attributable to an overestimation of $k_{on}$ rates during the BLI experiments due to rebinding of virus, which is often a caveat associated with bulk measurements[26,28].

**Binding of reovirus σ1 attachment protein to sialic acid alters β1 integrin interaction.** We previously showed that α-SA binding by the reovirus σ1 attachment protein enhances its avidity for JAM-A[18]. Based on this finding, we tested whether α-SA binding also alters reovirus interactions with β1 integrin. In these experiments, we studied β1 integrin binding in the presence of $Mn^{2+}$ to (i) virions of T3SA-, a glycan-blind reovirus strain containing a P204L point mutation in the α-SA-binding site of σ1, and (ii) virions of T3SA+, in the presence of acetylneuraminic acid (Neu5Ac), a short α-sialylated glycan. Binding of α-SA by σ1 triggers an extended conformation (Fig. 1b) similar to the σ1 conformation in ISVPs (Fig. 1c)[11].

AFM experiments revealed that T3SA- virions bound β1 integrin at a significantly lower frequency (7 ± 1%) (mean ± S.D., $N = 11264$) relative to T3SA+ virions (Fig. 4b), while both interactions displayed similar kinetics (a single energy barrier with $x_u = 0.50 ± 0.04$ nm and $k_{off} = 0.58 ± 0.2\ s^{-1}$ for T3SA-) (Fig. 4c, d). T3SA- interactions with integrin displayed a smaller $x_u$ (distance to the transition state) than T3SA+, indicating that the T3SA- energy landscape has a narrower valley that can accommodate less conformational variability. The narrower energy valley and the lower BF suggest that the single point mutation in T3SA- σ1 leads to a more rigid or compact conformation of the protein, which in turn diminishes accessibility to β1 integrin by the putative integrin-binding capsid component, λ2 (Fig. 1b). To test this hypothesis, we quantified the binding probability of each strain as a function of contact time (Fig. 4e). Dissociation constants determined using this approach were similar for both virus strains ($K_D = 103 ± 44$ nM

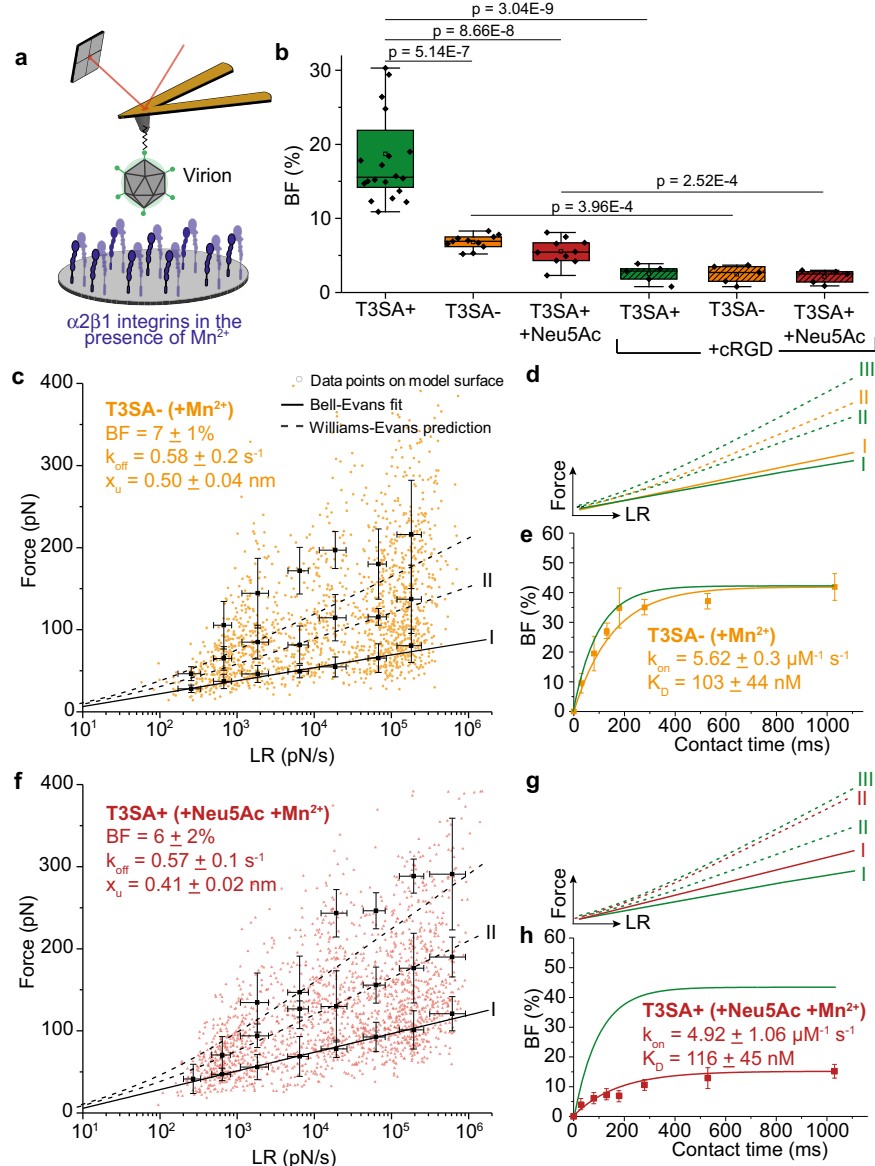

**Fig. 4 Binding of the reovirus σ1 attachment protein to sialic acid alters β1 integrin interaction.** The effect of a point mutation in the SA-binding site of σ1 protein (**c–e**) or addition of exogenous α-SA (**f–h**) on reovirus-α2β1 integrin interactions was determined using model surfaces. **a** Schematic depicts the experimental set up. All experiments were conducted in the presence of $Mn^{2+}$. **b** Box plot shows BF between T3SA+ or T3SA- virions and integrins quantified using AFM under the conditions shown. The horizontal line within the box indicates the median, boundaries of the box indicate the 25th and 75th percentile, and whiskers indicate the highest and lowest values. An open square within each box indicates the mean. **c** DFS plot shows interaction forces between T3SA- and integrins. **d** Comparison of forces required to rupture bonds between integrins and T3SA+ (green) or T3SA- (yellow). **e** BF plotted as a function of contact time shows comparable $k_{on}$ and $K_D$ values for T3SA+ (green) and T3SA- (yellow) interactions with integrins. **f–h** DFS plot (**f**) and kinetic on-rate measurements (**h**) show differences in T3SA+ interactions with β1 integrins in the presence (red) and absence (green) of α-SA (1 mM Neu5Ac) (**g**). Error bars indicate s.d. of the mean. All data are representative of N = 5 independent experiments. P values were determined by two-sample t-test using Origin.

**Table 2 Comparison of kinetic and thermodynamic parameters describing sialic acid dependence of reovirus virion interaction with β1 integrin.**

| $Mn^{2+}$ | T3SA+ | T3SA- | T3SA+ +Neu5Ac |
|---|---|---|---|
| BF [%] | 19 ± 7 | 7 ± 1 | 6 ± 2 |
| $k_{off}$ [$s^{-1}$] | 0.4 ± 0.2 | 0.58 ± 0.2 | 0.57 ± 0.1 |
| $k_{on}$ [$\mu M^{-1}\ s^{-1}$] | 5.83 ± 0.6 | 5.62 ± 0.3 | 4.92 ± 1.1 |
| $K_D$ [nM] | 68 ± 56 | 103 ± 44 | 116 ± 45 |
| $x_u$ [nm] | 0.69 ± 0.07 | 0.50 ± 0.04 | 0.41 ± 0.02 |

for T3SA- and 68 ± 56 nM for T3SA+ interaction with β1 integrin). These results suggest that differences in the interaction of T3SA+ and T3SA- virions with integrin are due to differences in the accessibility of the integrin-binding site in virions and not due to changes in affinity (parameters summarized in Table 2). Experiments conducted using living Lec2 cells confirmed these findings in a more physiological context (Supplementary Fig. 4). In addition, injection of cRGD reduced the BF significantly, confirming the specificity of T3SA- binding to β1 integrin (Fig. 4b).

We next investigated the influence of α-SA (Neu5Ac) binding on reovirus interaction with integrin (Fig. 4b, f–h). The addition

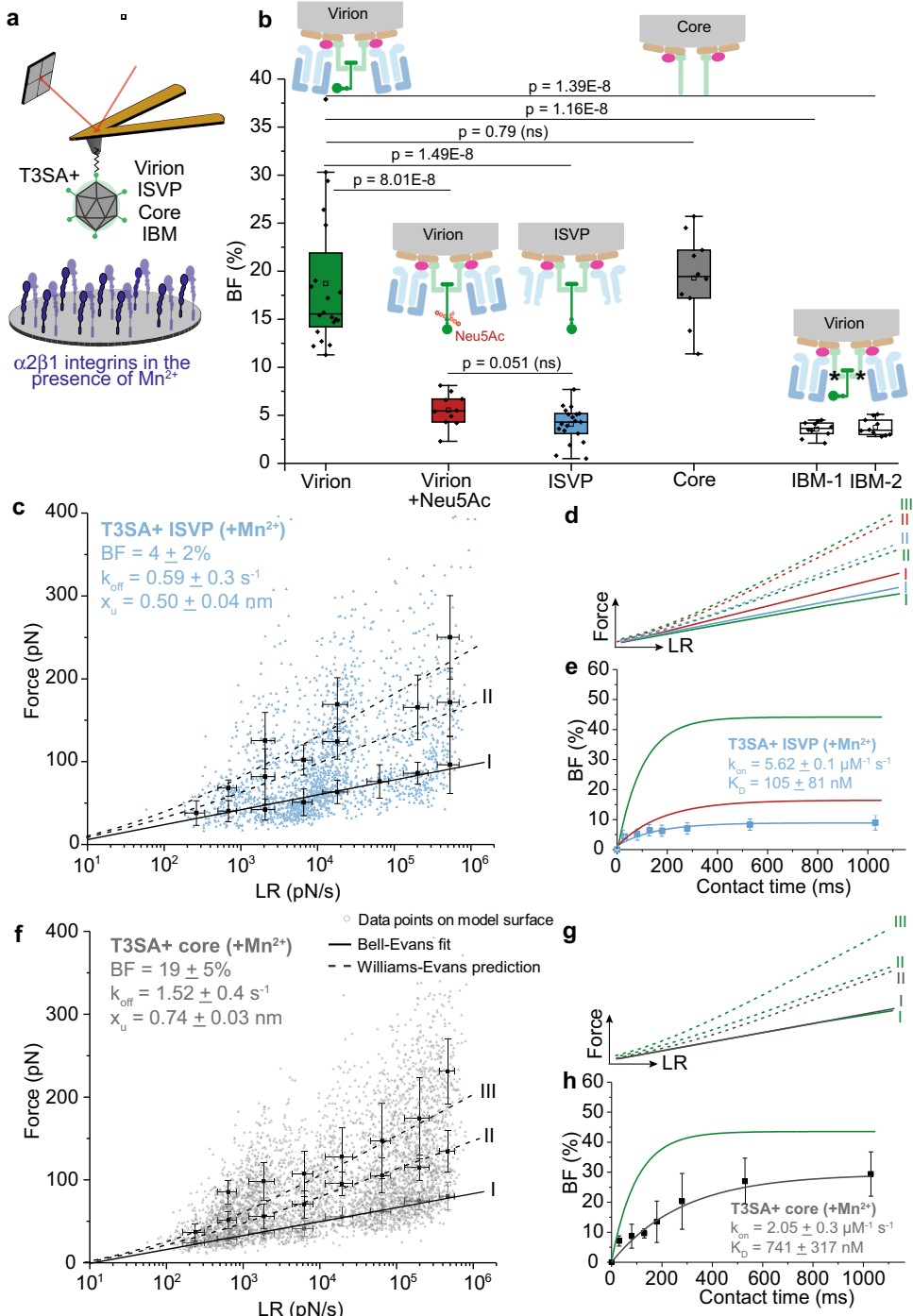

**Fig. 5 Analysis of β1 integrin interactions with reovirus disassembly intermediates suggests λ2 as the viral integrin ligand. a** Schematic depicts the experimental set up. All experiments were conducted in the presence of Mn²⁺. **b** Box plot of BF between β1 integrins and T3SA+ virions, ISVPs, cores, or IBM-1/-2 (integrin binding motif mutant) quantified using AFM under the conditions shown. The horizontal line within the box indicates the median, boundaries of the box indicate the 25th and 75th percentile, and whiskers indicate the highest and lowest values. An open square within each box indicates the mean. **c–e** Binding of T3SA+ ISVPs to β1 integrins (colored in blue) in comparison with virions before (colored in green) and after (colored in red) incubation with Neu5Ac. **c** DFS plot of the interaction forces between integrins and T3SA+ ISVPs show that ISVPs display a different kinetic bond rupture behavior compared with T3SA+ virions in the absence of Neu5Ac but the same behavior as T3SA+ virions in the presence of Neu5Ac (as shown in **d**). **e** $k_{on}$ and $K_D$ determined from measuring BF as a function of contact time indicates that the affinity of ISVPs for integrin is comparable to that of virions in the presence of Neu5Ac. **f–h** Binding of T3SA+ cores (colored in gray) to β1 integrins differs from that of T3SA+ virions (colored in green). Plots display DFS data (**f**), comparison of T3SA+ cores (gray) with virions (green) (**g**), and kinetic on-rate measurements (**h**). Control experiments are shown in Supplementary Fig. 5. Error bars indicate s.d. of the mean. All data are representative of $N = 3$ independent experiments. Ten AFM tips were analyzed per condition. *P* values were determined by two-sample *t*-test using Origin .

**Table 3 Comparison of kinetic and thermodynamic parameters describing β1 integrin interaction with T3SA+ reovirus disassembly intermediates.**

| $Mn^{2+}$ | Virion | Virion + Neu5Ac | ISVP | CORE |
|---|---|---|---|---|
| BF [%] | 19 ± 7 | 6 ± 2 | 4 ± 2 | 19 ± 5 |
| $k_{off}$ [$s^{-1}$] | 0.4 ± 0.2 | 0.57 ± 0.1 | 0.59 ± 0.3 | 1.52 ± 0.4 |
| $k_{on}$ [$\mu M^{-1}$ $s^{-1}$] | 5.83 ± 0.6 | 4.92 ± 1.06 | 5.62 ± 0.1 | 2.05 ± 0.3 |
| $K_D$ [nM] | 68 ± 56 | 116 ± 45 | 105 ± 81 | 741 ± 317 |
| $x_u$ [nm] | 0.69 ± 0.07 | 0.41 ± 0.02 | 0.50 ± 0.04 | 0.74 ± 0.03 |

of Neu5Ac led to a strong decrease in the BF of T3SA+ virions to integrin coated on a model surface (6 ± 2%, $N = 10240$ in the presence of Neu5Ac) (Fig. 4b). The parameters $k_{off}$, $k_{on}$, and the resulting $K_D$ remained unchanged (Fig. 4f, h and Table 2), and $x_u$ decreased to 0.41 ± 0.02 nm after the addition of Neu5Ac. The decrease in BF is likely due to a Neu5Ac-induced extension in σ1 conformation[11] (Fig. 1b), which may hinder accessibility of λ2 to integrin, thus preventing efficient interaction. These results suggest that the σ1 conformation influences the accessibility to the integrin-binding site but not the binding pocket itself.

**Accessibility of λ2 reovirus protein is key to integrin binding.** Our results thus demonstrate that reovirus can directly bind to β1 integrin and suggest a potential role of σ1 in influencing binding site accessibility. We next investigated whether the reovirus λ2 capsid protein is required for β1 integrin binding. We extended our studies from T3SA+ virions to the disassembly intermediates, ISVPs and cores (Fig. 1c, d). Among capsid components exposed on the reovirus surface, only λ2 is common to all three particle types (virions, ISVPs, and cores), albeit with variable accessibility. We therefore hypothesized that if λ2 mediates reovirus interactions with integrin, all three particle types should be capable of engaging integrin, although with different probabilities. To test this hypothesis, we investigated the interaction of all three particle types with integrins coated on a model surface (Fig. 5a). When compared with virions, ISVPs bound to β1 integrins with similar affinity, a small decrease in $x_u$, and a significant decrease in the BF ($P < 0.0001$) (Fig. 5b–e). The BF of ISVPs is similar to that of virions in presence of Neu5Ac. These observations suggest that an extended conformation of σ1 in ISVPs hinders efficient binding to integrins.

Reovirus cores contain λ2 and lack all outer-capsid proteins (μ1, σ3, and σ1)[8,20] (Fig. 1e). The BF of T3SA + cores to β1 integrin (19 ± 5%, $N = 10240$) (Fig. 5b) is higher than that of ISVPs and similar to virions. Cores also display a similar $x_u$ as virions (Fig. 5f). However, $k_{off}$, $k_{on}$, and consequently $K_D$ differ, indicating that the β1 integrin-core interaction is less stable than β1 integrin-virion interaction (Fig. 5f–h). Reduced stability of the β1 integrin-core complex may be attributable to differences in λ2 conformation in virions and cores[20]. Furthermore, the diminished stability of the β1 integrin-core complex is reasonable, as cores are produced during disassembly following internalization and do not need to interact with integrin. Thus, these results suggest that reovirus binds β1 integrins using its λ2 capsid protein.

Studies of β1 integrin interaction with virions, ISVPs, and cores were also performed in the presence of Neu5Ac or lacto-N-neotetraose (LNnT), a non-α-sialylated glycan. We observed, as anticipated, that for virions the injection of Neu5Ac but not LNnT is associated with a significant decrease in the BF, compatible with a change in σ1 conformation triggered by the α-sialylated glycan. For ISVPs and cores, no change was observed,

which is in good agreement with the fact that σ1 is in an extended conformation on ISVPs and absent on cores (Supplementary Fig. 5 and Table 3).

To test whether the putative IBMs, RGD and KGE, in λ2 are required for reovirus interactions with integrin, we altered these motifs and recovered T3SA+ IBM mutant strains. The RGD/KGE motifs in the λ2 protein were mutated to RGA/KGA (T3SA+ IBM-1) or TGA/TGA (T3SA+ IBM-2). The BF of the mutants to β1 integrin decreased to 4 ± 1% ($N = 10240$) and 4 ± 1% ($N = 10240$) for T3SA+ IBM-1 and T3SA+ IBM-2, respectively (Fig. 5b, white boxes). This substantial reduction in the binding of the IBM mutants to integrins, in comparison with wildtype T3SA+ (19 ± 5%, $N = 10$), unambiguously demonstrates that the RGD and KGE motifs in λ2 are required for engaging β1 integrin.

**Reovirus binding to β1 integrins activates clathrin-mediated internalization.** We investigated whether binding of reovirus virions to β1 integrin has a functional role in triggering clathrin-mediated uptake into host cells, as the NPXY motifs in β1 integrins have known functions in recruiting adaptor proteins and clathrin. To monitor the dynamics of reovirus binding and clathrin recruitment, we used our recently developed approach[29] in which virus-bound nanoparticles (NPs) are coupled with fluidic force microscopy (Fluid-FM) and a fast-scanning confocal microscope on living cells (Fig. 6a)[29]. A gold NP functionalized with fluorescently labeled T3SA+ particles was trapped at the apex of the Fluid-FM probe and subsequently brought in contact with CHO cells expressing JAM-A and blue fluorescent protein (BFP)-tagged clathrin. Virus-coated NPs were lowered onto the cell surface in a force-controlled manner. A significant increase in BFP signal corresponding to clathrin recruitment was observed at the plasma membrane surrounding T3SA+ virion-coated NPs within ~50 s after contact (Fig. 6b–d, green colored data; Supplementary Movie 1). Injection of cRGD and KGE peptides, which block β1 integrin binding, or Neu5Ac, which induces σ1 extension, abolished the recruitment of clathrin to the plasma membrane (Fig. 6c, e, black, red, and dashed colored data respectively; Supplementary Fig. 6a, b and Supplementary Movies 2, 3). These data indicate that efficient virus binding to β1 integrin is required to trigger clathrin recruitment and subsequent endocytosis. Accordingly, NPs functionalized with T3SA+ ISVPs, which poorly interact with β1 integrin (low BF), did not recruit clathrin (Fig. 6d, e, blue colored data; Supplementary Fig. 6c, Supplementary Movie 4). These results are also in line with the use of an alternative endocytic pathway by ISVPs, likely mediated by caveolin[23]. Similarly, NPs decorated with T3SA+ IBM-1 or T3SA+ IBM-2 were unable to recruit clathrin (Fig. 6d, e, black and gray colored data; Supplementary Fig. 7a and b, Supplementary Movies 5 and 6). Of note, NPs functionalized only with Alexa 700 dyes also did not lead to clathrin recruitment (Fig. 6d, e, purple colored data; Supplementary Fig. 7c, Supplementary Movie 7). Thus, reovirus T3SA+ binding to β1 integrins via RGD/KGE motifs in the λ2 outer-capsid protein activates an internalization pathway by inducing rapid and specific recruitment of clathrin to the plasma membrane.

## Discussion

Attachment to cell-surface sialylated glycans and JAM-A mediates reovirus internalization. However, molecular mechanisms underlying reovirus uptake are poorly understood. Truncation of the JAM-A cytoplasmic tail does not impede reovirus cell entry[22], suggesting that other host factors mediate intracellular signaling to trigger internalization. Several lines of evidence point to a pivotal role for β1 integrin in receptor-mediated reovirus endocytosis. Viruses belonging to diverse families such as herpes

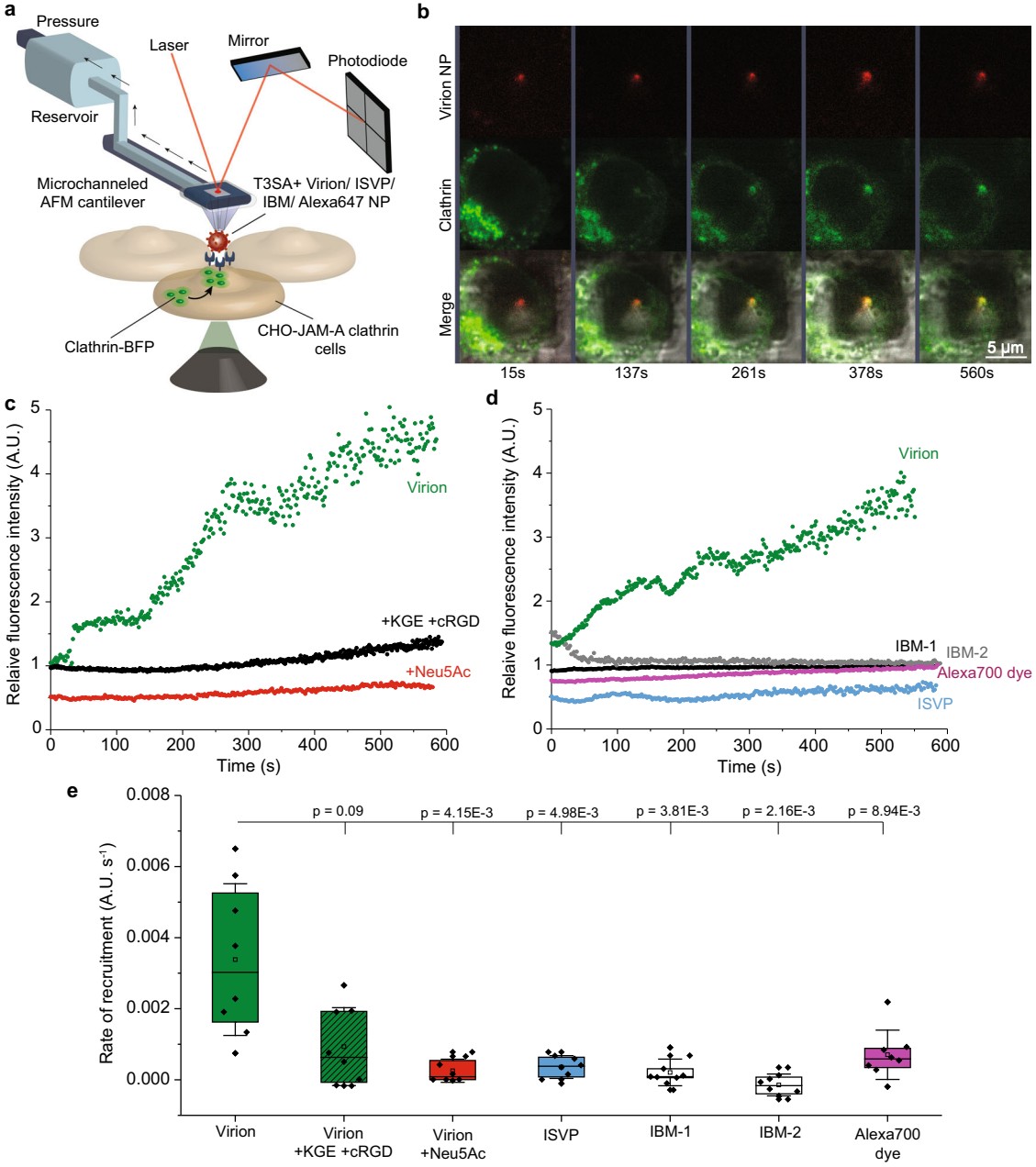

**Fig. 6 Integrins function in clathrin recruitment by reovirus bound to cells.** Integrin-induced recruitment of clathrin (labelled with blue fluorescent protein [BFP]) at the plasma membrane following reovirus binding was quantified using Fluid-FM coupled with confocal microscopy. **a** Fluid-FM coupled to a confocal imaging setup. Fluorescent T3SA+ reovirus particles covalently coupled to gold-coated nanoparticles were trapped using a micro-channeled cantilever and brought in contact with clathrin-BFP-expressing CHO-JAM-A cells. Fast-scanning confocal microscopy was used to simultaneously visualize reovirus and clathrin fluorescence signals. **b** Representative time-lapse images from fluid-FM/confocal imaging show recruitment of clathrin (BFP, for better visibility shown in green instead of blue) to a reovirus (Alexa 647, red) contact site. **c** Recruitment of clathrin observed over time using blue fluorescence intensity around beads coated with T3SA+ virions in the absence (green) or presence of integrin-blocking peptides KGE and cRGD (black) or Neu5Ac (red). **d** Recruitment of clathrin observed over time using blue fluorescence intensity around beads coated with T3SA+ virions (green) and control particles ISVP (blue), IBM-1 (black), IBM-2 (gray), or Alexa 700 dye (purple). Representative images from control experiments are displayed in Supplementary Figs. 6 and 7. **e** Box plot shows the rate of clathrin recruitment calculated from the quantification of fluorescence over time (slopes of intensity vs. time curves fit to linear regressions). The horizontal line within the box indicates the median, boundaries of the box indicate the 25th and 75th percentile, and whiskers indicate the highest and lowest values. An open square within each box indicates the mean. Error bars indicate s.d. of the mean value. All data are representative of $N = 10$ independent experiments. $P$ values were determined by two-sample $t$-test using Origin.

virus[30], hantavirus[31], and echovirus[32], among others use integrins for entry. This underscores the importance of understanding the mechanism of integrin interaction to gain insights into the initial steps in virus interaction with a host cell to establish infection. Of particular note is adenovirus, which shares similarities with

reovirus in structures of attachment proteins and receptors[14] and also undergoes integrin-mediated internalization[33]. The reovirus λ2 protein contains conserved IBMs, RGD and KGE[21,34], on surface-exposed loops, which are well-positioned to interact with integrin. While both reovirus virions and ISVPs contain these

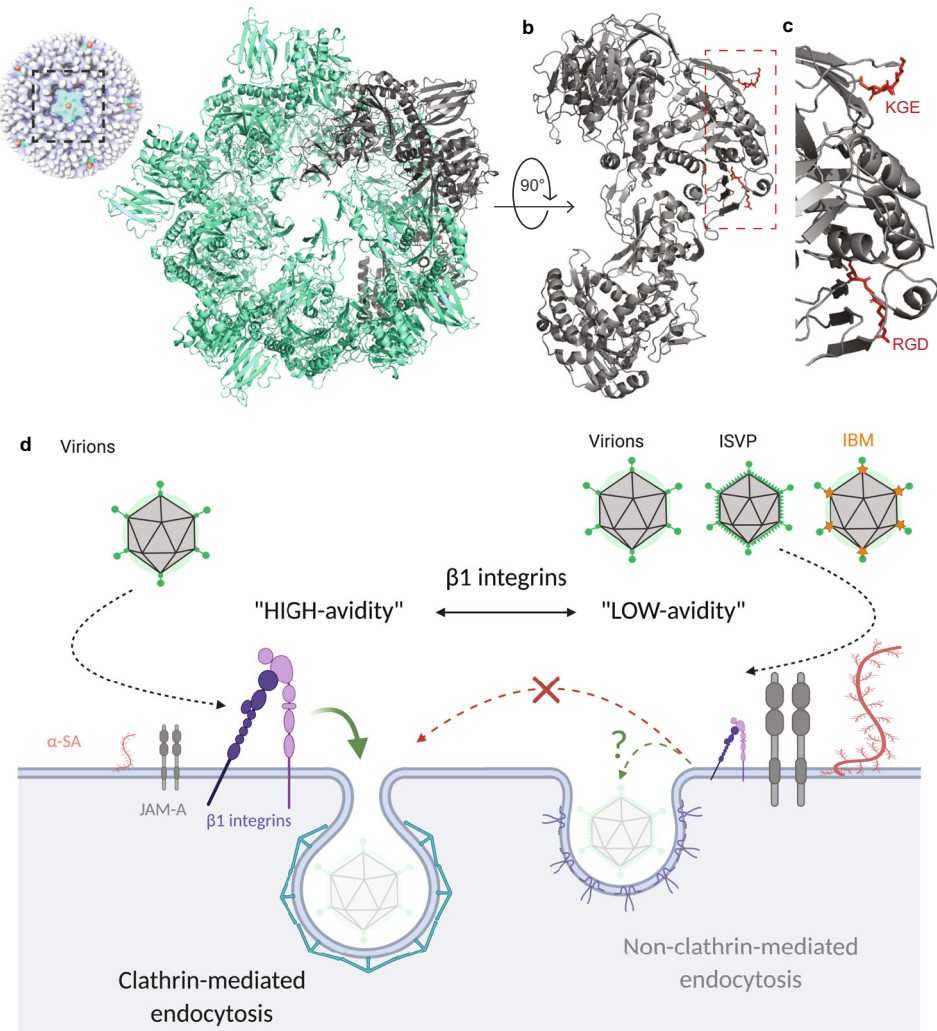

**Fig. 7 Integrin interaction with λ2 reovirus outer capsid protein mediates clathrin-recruitment for entry into host cells. a** Crystal structure of the λ2 turret (pentamer, about 120 Å diameter and 80 Å tall), viewed from the top. Cryo-electron microscopy reconstruction of the virion is shown on the side[8,55]. The five elongated λ2 monomers, four shown in green and one in gray, wrap around the other surface, with their long axes about 45° to the radial direction. **b** The λ2 monomer, viewed from the side (the gray monomer in **a**). β1 integrin binding sequences, RGD and KGE, are highlighted as red sticks. **c** Zoom-in on β1 integrin binding domain in λ2. Images are based on related crystal structure 1EJ6 [https://doi.org/10.1038/35010041]. **d** Conclusive model: Virions efficiently bind β1 integrins ("high-avidity" binding), which mediates recruitment of clathrin for virus internalization. Virions in the presence of abundant free α-SA or ISVPs bind poorly to integrins ("low-avidity" binding), potentially because of steric hindrance from conformational changes in σ1. Consequently, they use alternate mechanisms for entry such as caveolin-mediated endocytosis, similar to virions lacking the β1 integrin domain (IBM).

motifs, blocking β1 integrin with antibodies diminishes infection by reovirus virions but not ISVPs[22]. These data suggest a model in which λ2 protein interactions with β1 integrin mediate endocytosis of reovirus virions.

In this study, we used AFM in combination with confocal microscopy to demonstrate direct interactions between reovirus and β1 integrin. Using single-virus force spectroscopy, we quantified the binding strength and extracted kinetic and thermodynamic parameters characterizing single bonds established between reovirus and β1 integrin. As the conformation and functional state of integrins are regulated by divalent cations, we also tested the influence of $Ca^{2+}$, $Mg^{2+}$, and $Mn^{2+}$ ions on the reovirus-integrin interaction. Using these strategies, we discovered that reovirus virions specifically bind to β1 integrin with high affinity in a cation-dependent manner. While little specific interaction is observed between reovirus and β1 integrins in the presence of $Ca^{2+}$, the interaction is favored in the presence of $Mg^{2+}$ with a $K_D$ of ~548 nM and is maximal in the presence of $Mn^{2+}$, reaching a $K_D$ of ~68 nM. These observations perfectly

correspond to regulation of β1 integrin ligand-binding capacity by various divalent cations and highlight the physiological relevance of the results reported here. In addition to altering integrin conformation, it is possible that changes in cation concentration influence other steps in reovirus entry, e.g., affinity of the virus for other host factors required for internalization and sorting in the endocytic pathway, capsid disassembly, or membrane penetration. However, a significant reduction of cation-mediated effects on reovirus binding and internalization by cRGD suggests that the effects of cation on reovirus internalization are likely mediated through induced conformation changes in integrin. We also compared the capacity of the different in vitro-generated reovirus disassembly intermediates with known structural composition to bind β1 integrins. Comparison of integrin-binding by virions and ISVPs did not reveal differences in $k_{on}$ and $k_{off}$ but showed a lower BF of ISVPs compared with virions. However, the BF of virions decreased in presence of Neu5Ac and became similar to that of ISVPs. These results, along with a higher $x_u$ for virions compared with ISVPs, suggest that the accessibility of the

integrin-binding site is sterically hindered in ISVPs and in virions treated with Neu5Ac, which correlates with an extension of σ1[11]. Reovirus cores, which lack σ1, bind integrin with a high probability, which agrees with enhanced access to λ2 on the surface of cores. However, the kinetic and thermodynamic parameters reveal a less stable interaction of cores with integrin, which may be due to conformational changes in λ2 that occur during conversion of virions and ISVPs to cores. Indeed, structural data indicate that while the surface-exposed flaps in the pentameric λ2 turrets exist in a closed conformation in virions, the flap domains open outward in cores. These conformational changes in λ2 could explain the change in integrin-binding by cores, yielding a less stable and shorter-lived complex relative to virions. Together, these results provide evidence that reovirus virions bind β1 integrins more efficiently than ISVPs and cores, particle types in which access to the λ2 protein is sterically hindered. β1 integrin binding experiments carried out with T3SA+ mutants lacking the λ2 RGD/KGE IBMs demonstrate conclusively that those motifs in the λ2 outer-capsid protein are required for integrin binding by reovirus (Fig. 7a–c). Conclusions drawn from a structural point of view await high-resolution structural analyses for conformation. These results explain previously observed differences in the uptake mechanisms used by virions and ISVPs following receptor engagement[35,36]. In summary, virions that efficiently bind β1 integrins (through "high-avidity" binding), induce a recruitment of clathrin, which likely promotes virus internalization. On the other hand, ISVPs bind poorly to integrins (through a "low-avidity" binding), potentially because of steric hindrance resulting from an σ1 extended conformer. Consequently, they probably use an alternate mechanism for entry such as caveolin-mediated endocytosis, similar to virions lacking the β1 integrin domain (Fig. 7d).

Findings reported here shed new light on the complex interplay of interactions between reovirus proteins and β1 integrin in promoting virus entry. A wide variety of viruses, including adenovirus, coronavirus, picornavirus, and reovirus, have adopted similar mechanisms to gain entry into host cells. A better understanding of these mechanisms will illuminate commonalities in underlying pathogen-host interactions that may provide valuable insights for the development of broad-spectrum antiviral therapeutics. In addition, these findings might contribute to the development of more effective virus-based oncolytic treatments by enhancing the targeting of cancer cells.

## Methods

**Propagation of T3SA+ and T3SA− reovirus stocks.** Stocks of reovirus strains T3SA+ and T3SA− were prepared by plaque purification and passaging the viruses 3–4 times in L929 cells. Purified virions were prepared from infected L929 cell lysates by cesium chloride gradient centrifugation as described[37]. Infected cells were lysed by sonication, and virions were extracted from lysates using vertrel-XF and further layered onto 1.2–1.4 g/cm3 caesium chloride step gradients and centrifuged at 87,000 × g at 4 °C for 18 h. The band corresponding to the density of reovirus particles (~1.36 g/cm3) was collected and exhaustively dialyzed against virion-storage buffer (150 mM NaCl, 15 mM MgCl2 and 10 mM Tris [pH 7.4]). T3SA+ differs from T3SA- by a single point mutation in σ1 (P204L) that abolishes the capacity of the virus to bind sialic acid. Virus particle concentration was determined from optical density at 260 nm (1 $OD_{260}$ = 2.1 × 10^{12} particles mL^{−1})[38]. ISVPs and cores were prepared by incubation of virions with 2 mg mL^{−1} α-chymotrypsin (Sigma-Aldrich) at 37 °C for 60 min and 90 min, respectively[39,40]. ISVPs were used immediately in experiments, or ISVPs and cores were further purified using cesium chloride gradient centrifugation. Purified reovirus virions or ISVPs were labeled with succinimidyl esters of Alexa Fluor 488 or Alexa Fluor 647 (Invitrogen) to prepare fluorescently-labeled particles[41].

**Reovirus λ2 mutants used in this study.** *Engineering λ2 mutants.* Putative integrin-binding motifs (IBMs), RGD and KGE, in reovirus protein λ2 were changed to RGA and KGA or TGA and TGA, respectively. The mutations were engineered into the plasmid pT7-L2T1L (Addgene 33287) using PCR-based amplification of two linear fragments: one fragment encoding the region between RGD and KGE motifs of the *L2* gene, which encodes λ2, and a second overlapping fragment encoding the remainder of the plasmid. Mutations to the RGD and KGE motifs were designed within the PCR primers listed in Supplementary Table 1. The two PCR fragments were ligated using Gibson assembly (NEB, E2611S) according to the manufacturer's instructions.

*Recovery and characterization of λ2 mutant stocks.* T3SA + containing mutations in the IBMs were recovered using plasmid-based reverse genetics[42,43]. The T3SA + (λ2: RGA/ KGA) mutant is referred as T3SA + IBM-1 and the T3SA + (λ2: TGA/TGA) mutant as T3SA + IBM-2. Eight plasmids encoding reovirus gene segments (pT7-L1-M2T1L [Addgene, 33303], pT7-L3-S3T1L [Addgene, 33305], pT7-M1T1L [Addgene, 33289], pT7-M3T1L [Addgene, 33291], pT7-S2T1L [Addgene, 33293], pT7-S4T1L [Addgene, 33295], pT7-S1T3SA + [Genbank accession no. EF494441] or pT7-S1T3SA-[44] and an *L2* plasmid encoding mutations) were co-transfected into BHK-T7 cells. All reovirus genes are derived from strain T1L except for S1, which is derived from the strain T3C44-MA[44,45].

Viruses were plaque-purified and propagated in L929 cells for three passages. Infected cells were lysed by sonication, and virions were extracted from lysates using vertrel-XF[37,46]. The extracted virions were layered onto 1.2–1.4 g cm^{−3} cesium chloride step gradients and centrifuged at 87,000 × g at 4 °C for 18 h. The band corresponding to the density of reovirus particles (~1.36 g cm^{−3}) was collected and dialyzed exhaustively against virion-storage buffer (150 mM NaCl, 15 mM MgCl2, and 10 mM Tris [pH 7.4]). Mutations in the L2 gene of purified viruses were confirmed by RT-PCR (Qiagen, 210210) of the L2 gene (*T1L-L2-2502-fwd, 5′CGGTCCTGAAGCGAAGATTC; T1L-L2-3901-rev, 5′GCTCACGAGGGGTAATGAGTTAC*) and Sanger sequencing (*T1L-L2-3378-rev, 5′TGCATCTGTTATGTCCACATCC*) (Supplementary Table 1).

**Cell lines used in this study.** *Culture of CHO cells.* Engineering and characterization of CHO cells expressing JAM-A has been described[11]. CHO-JAM-A cells stably express a puromycin-resistance gene and JAM-A. Cells were grown in Ham's F12 medium (Sigma-Aldrich) supplemented to contain 10% fetal bovine serum (FBS), penicillin (100 U ml^{−1}), and streptomycin (100 µg mL^{−1}) (Invitrogen) at 37 °C in a humidified atmosphere with 5% CO2. During alternate passages, 20 µg mL^{−1} puromycin was added to the medium.

*Culture of Lec2 cells.* Lec2 cells (ATCC, #CRL-1736) were grown in Mem α, nucleosides medium (Gibco) supplemented to contain 10% FBS, penicillin (100 U mL^{−1}), and streptomycin (100 µg mL^{−1}) at 37 °C in a humidified atmosphere with 5% CO2.

*Transient transfection of CHO-JAM-A cells with clathrin BFP (CHO-JAM-A mTagBFP2-clathrin).* CHO-JAM-A cells were transfected with a plasmid expressing N-terminal mTagBFP2-tagged clathrin light chain (pEGFP mTagBFP2-clathrin light chain) using Lipofectamine LTX (Invitrogen) according to the manufacturer's protocol. The GFP was replaced with mTagBFP2 in the pEGFP plasmid in the following manner. The sequence of mTagBFP2 was transferred from mTagBFP2-ER-5 (Addgene #55294) and further cleaved with NheI and EcoRI, which replaced the sequence of GFP in pEGFP-GFP11-Clathrin light chain (Addgene #70217) with same restriction sites. Since there was a termination codon in the cleaved sequence, one base pair was removed. Cells were used in fluid-FM experiments 48 h post-transfection.

**Functionalization of AFM tips.** Single-virus particles were grafted onto the AFM tip (MSCT and PFQNM-LC probes, Bruker) surface using ethanolamine coating and NHS-PEG27-acetal linker following a three-step protocol[47].

*Aminofunctionalization.* The tips were first washed with chloroform (2 × 10 min), rinsed with ethanol, dried under a gentle stream of nitrogen gas, and cleaned for 10 min using an ultraviolet radiation and ozone (UV-O) cleaner (Jetlight). The tips were then incubated overnight in freshly prepared ethanolamine solution (3.3 g of ethanolamine hydrochloride dissolved in 6.6 mL of dimethylsulfoxide [DMSO]).

*Linker attachment.* The amino-functionalized tips were washed three times with DMSO, two times with ethanol (1 min each), and dried with nitrogen, while 1 mg of acetal-PEG27-NHS heterobifunctional linker was dissolved in 0.5 mL of chloroform. The tips were immersed in the linker solution and 30 µL of triethylamine was added to catalyse the attachment reaction. This protocol yields a low grafting density of the linker on the AFM tip and ensures coupling of no more than one (in most cases) virus particle per tip, as observed using confocal microscopy (Supplementary Fig. 1b)[24]. After 2 h of incubation, the tips were washed three times with chloroform, dried with nitrogen gas, and stored under argon until the next day.

*Virus coupling.* After immersion of the tips for 10 min in 1% citric acid in milliQ water to allow conversion of the chemically inert acetal into an active aldehyde, the tips were washed three times with milliQ water, dried with nitrogen, and coupled to the virus. The tips were immersed in 100 µL of virus solution (~10^8–10^9 particles mL^{−1} in virion-storage buffer) and 2 µL of a freshly prepared solution of NaCNBH3 (~6% wt. vol^{−1} in 0.1 M NaOH(aq)) for 1 h. The virus-functionalized tips were not allowed to dry, and tips were transferred rapidly (<20 s) ensuring that a drop of buffer remained on the cantilever and tip. The coupling reaction was quenched by adding and gently mixing 5 µL of 1 M ethanolamine solution (pH 8). After 10 min, the tips were rinsed three times with ice-cold phosphate buffered saline (PBS) and stored in virion-storage buffer within an icebox until use. Experiments were conducted on the same day following virus coupling.

**Preparation of β1 integrin-coated model surfaces**. α2β1 integrins were grafted on gold-coated silicon substrates using N-hydroxysuccinimide (NHS)/1-ethyl-3-(3-dimethylaminopropyl)carbodiimide (EDC) chemistry in two steps.

*COOH functionalization.* Gold-coated silicon substrates were washed with absolute ethanol (for 5 min), dried with nitrogen gas, and cleaned for 15 min using the UV-O cleaner. Surfaces were then chemically activated by overnight immersion in alkanethiol solution (99% 11-mercapto-1-undecanol 1 mM [inactive] and 1% 16-mercaptohexadecanoic 1 mM [active] in absolute ethanol) (Sigma-Aldrich).

*Protein coupling.* Activated substrates were rinsed with ethanol and dried with nitrogen gas, before immersion for 30 min in a solution containing 10 mg mL$^{-1}$ NHS (Sigma-Aldrich) and 25 mg mL$^{-1}$ EDC (Sigma-Aldrich) (in milliQ water). Substrates were then rinsed with milliQ water and incubated for 1 h with 50 µL of α2β1 integrins (100 µg mL$^{-1}$ in PBS, R&D Systems). Finally, surfaces were washed ten times with PBS while never being allowed to dry and used in experiments on the same day following preparation.

Integrin-coated surfaces remained stable and homogenous during repeated scans and had a thickness of ~3.0 ± 0.5 nm measured using a scratching experiment (Supplementary Fig. 1a). A typical scratching experiment involves imaging of a small area (500 nm × 500 nm) on the model surface at high forces with a stiff cantilever, which results in scratching of the attached biomolecules. A larger area (3 µm × 3 µm) is then scanned at a lower force, which allows measurement of thickness of the scratched hole, corresponding to the thickness of the deposited integrin layer in this case.

**FD-based AFM on model surfaces**. AFM Nanoscope Multimode 8 (Bruker) was used (Nanoscope software v9.2) to conduct FD-based AFM. Virus-functionalized MSCT-D probes or MSCT-E (with spring constants calculated using thermal tune[48], ranging from 0.024 to 0.043 N m$^{-1}$ or 0.098 to 0.124 N m$^{-1}$ respectively) were used to record force curves from 5 × 5 µm model surface areas in the force-volume (contact) mode.

*Dynamic force spectroscopy (DFS) experiments (to measure $k_{off}$ and $x_u$).* The approach velocity of the AFM tip was kept constant at 1 µm s$^{-1}$, and retraction velocities were varied from 0.1, 0.2, 1, 5, 10 to 20 µm s$^{-1}$ (MSCT-D probes) and 20–30 µm s$^{-1}$ (MSCT-E probes, fast FV) to probe the energy landscape of the β1 integrin-reovirus interaction over a wide range of LRs (i.e., the load applied on the bond over time). The pulling velocity (v) and loading rate (LR) are related as follows: LR = ΔF/Δt = $k_{eff}$ · v, where ΔF/Δt is the applied force over time and $k_{eff}$ is the effective spring constant of the system. The ramp size was set to 500 nm and the maximum force was limited to 500 pN, with no surface delay. The sample was scanned using a line frequency of 1 Hz, and 32 pixels were scanned per line (32 lines in total with 1024 data points [FD curves] per retraction speed). All FD-based AFM measurements were obtained in virion-storage buffer at ~25 °C. Force curves were analyzed using the Nanoscope analysis software v1.9 (Bruker). To identify peaks corresponding to adhesion events between virus particles linked to the PEG spacer and the integrin-coated model surface, the retraction curve before bond rupture was fitted with the worm-like chain model for polymer extension[49]. The worm-like chain model describes the force-extension relationship for semi-flexible polymers and is defined by the following equation, where $l_P$ is the persistence length, $L_c$ the contour length, and $k_bT$ the thermal energy:

$$F = \frac{k_b T}{l_P} \left( \frac{1}{4\left(1 - \frac{x}{L_c}\right)^2} - \frac{x}{L_c} - 0.25 \right) \quad (1)$$

Origin 2009 software (OriginLab) was used to display the results as dynamic force spectroscopy (DFS) plots to fit histograms of rupture force distributions for distinct loading rate ranges and to apply various force spectroscopy models to extract the number of bonds mediating interaction between substrate and ligand, $k_{off}$, and $x_u$[24,25,50].

Briefly, to determine whether single or multiple bond rupture between virions and integrins are taking place, bond rupture forces were analyzed through distinct discrete ranges of LRs, plotted as force histograms and further fitted with multi-peak Gaussian distribution, as previously described[24,25,50]. Using these distributions, we determine the most probable unbinding force of each force peak (maximum of rupture force distribution) and overlay them on the DFS plot. Using the Bell–Evans model[26], we extracted the kinetic parameters of single bonds and used the predictive Williams-Evans model[27] to determine whether the multiple bonds are correlated or not.

*Contact time experiments (to measure $k_{on}$ and $K_D$).* The kinetic on-rate ($k_{on}$) of the interaction can be estimated on model surfaces by the following equation, assuming the interaction follows pseudo first order kinetics:

$$k_{on} = (c_{eff} \cdot \tau)^{-1} \quad (2)$$

where $c_{eff}$ is the effective concentration of free partners available for interaction at equilibrium [M] and τ is the interaction time [s]. $c_{eff}$ can be calculated as the number of binding sites (i.e., the valency of interaction measured on the corresponding DFS plot) over the effective volume accessible for free equilibrium interaction between the virus and integrins (i.e., a sphere with radius equal to the equilibrium length of the PEG linker and the size of the virus, ~88 nm). We extracted τ by fitting the BF over contact time (t) with the following equation using the Origin software:

$$BF = A \left[ 1 - \exp\left( \frac{-(t - t_0)}{\tau} \right) \right] \quad (3)$$

Where A is the maximum observable BF and $t_0$ is the lag time.

Experimentally, the BF between virus-functionalized tips and integrin-grafted model surfaces was measured for different hold times of 0, 50, 100, 150, 250, 500, and 1000 ms. All other parameters were kept constant: approach/reverse velocity: 1 µm s$^{-1}$, ramp size: 500 nm, force setpoint: 500 pN, line frequency: 1 Hz, and pixels scanned per line: 32[28,51]. $K_D$ was calculated subsequently as $k_{off}/k_{on}$.

**FD-based AFM on living Lec2 cells**. Experiments on living cells were performed using a Bioscope Resolve AFM (Bruker) operated in the PeakForce QNM mode (Nanoscope software v9.2) coupled to an inverted optical microscope (Zeiss Observer Z.1, 40x oil objective)[25,52]. The AFM was equipped with a 150 µm piezoelectric scanner and a cell culture chamber[24]. Experiments were conducted using a confluent layer of Lec2 cells seeded on 47 mm glass-bottomed Petri dishes (WillCo Wells). Cell integrity was maintained upon repeated scanning (Supplementary Fig. 2a–d). PFQNM-LC probes (Bruker, with 17 µm tip length, 65 nm tip radius, and 15° opening angle) were used to image 25 × 25 µm to 30 × 30 µm areas containing 2–4 flat, adjacent cells in contact. A "no-touch" calibration was performed by fixing the nominal spring constant of pre-calibrated cantilevers (ranging from 0.079 to 0.129 Nm$^{-1}$) and adjusting the sensitivity using thermal noise method. The AFM tip was oscillated in a sinusoidal manner with an amplitude of 750 nm and a frequency of 0.25 kHz, while the maximum force applied was limited to 500 pN. The sample was scanned at a 0.125 Hz frequency and 256 × 256-pixel resolution (i.e., 65536 FD curves per image). Individual FD curves displaying specific unbinding events were extracted and analyzed using Nanoscope analysis (v1.9, Bruker) and Origin 2009 (OriginLab) software. The baseline of retraction curves was adjusted using a linear fit on the last 30% of the curve. Further image processing and analysis were performed using software ImageJ (v1.52e) and Zen Blue 3.1 (Zeiss), respectively[24,25,50].

**Specific experimental setups**. *Monitoring the divalent cation dependent binding behavior of integrins to reovirus.* Divalent cation dependent behavior of integrins was studied in presence of 2 mM manganese (Mn$^{2+}$), 15 mM magnesium (Mg$^{2+}$, already present in the virion-storage buffer) or 2 mM calcium (Ca$^{2+}$) during in vitro experiments, and 0.5 mM Mn$^{2+}$ or Mg$^{2+}$ during measurements using living cells. Probing the same area with the same tip before and after injection of cations into the medium demonstrates that binding probabilities can be reliably compared between different conditions (Supplementary Fig. 2e–l).

*Monitoring effects of SA binding on integrin interaction.* To study potential influence of SA on integrin binding by reovirus, adhesion maps were recorded before and after addition of 1 mM N-acetylneuraminic acid (Neu5Ac, Sigma), an α-sialic acid terminated glycan that triggers σ1 conformational change[11]. Effect of another glycan lacking α-SA (lacto-N-neotetraose, LNnT), which is not supposed to bind to σ1, was also tested. The T3SA- mutant virus (lacking the SA binding capability due to a P204L mutation) was also probed both on model surfaces and living cells.

*Assessing the specificity of integrin-reovirus interactions.* The specificity of reovirus interaction with integrin was assessed by recording adhesion maps before and after injection of an integrin-blocking cRGD peptide (1 mg mL$^{-1}$, Sigma) in the presence of various divalent cations. cRGD blocks the RGD-binding site in integrins and thereby reovirus binding, leading to a dramatic decrease in the binding probability. Cells were fixed before addition of cRGD, as addition of this peptide led to cell detachment. The fixation did not substantially affect the binding behavior as shown in Supplementary Fig. 8. Cells were fixed using PBS containing 4% formaldehyde for 10 min (Invitrogen, Thermo Fisher Scientific) and washed with PBS.

*Localization of integrin binding site.* The location of integrin binding site in virions was further investigated by probing the interaction (as described before) of various disassembly intermediates (ISVP and core in addition to the viral particle) formed during the reovirus infection cycle, that differ in their structural glycoprotein composition and conformation. These interactions were also tested in the presence of 1 mM Neu5Ac or LNnT.

**Kinetic characterization of β1 integrin-reovirus interactions using bio-layer interferometry**. The cation dependency of reovirus binding to β1 integrins was validated using BLI as a complementary technique. The measurements were conducted at room temperature using a Blitz® device, amine reactive 2nd generation (AR2G) biosensors, and BLItz Pro™ software (v1.2, Pall ForteBio). After hydrating the biosensor for 10 min with PBS and running an equilibration step using acetate buffer (pH 4) (i.e., initial baseline), the surface of the biosensor was chemically activated by a 5 min incubation with freshly prepared solution of 20 mM EDC and 10 mM NHS in milliQ water. α2β1 integrins (5 µg mL$^{-1}$ in acetate buffer) were immobilized for 3 min, and the reaction was quenched by incubating the sensor tip in 1 M ethanolamine (pH 8) for 5 min. Before measuring the association of T3SA+ virus (~10$^{13}$ particles mL$^{-1}$ in virion-storage buffer; 5 min), another baseline was measured in virion-storage buffer. The virion-storage

buffer was supplemented to contain divalent cations (2 mM $Mn^{2+}$ or $Ca^{2+}$) to activate integrins. Finally, the dissociation was measured in virion-storage buffer supplemented to contain the same cation as for the baseline (5 min).

The sensorgrams (i.e., binding over time) obtained were processed and fitted to a nonlinear regression model using GraphPad (Prism 8). This model fits the association and dissociation steps to extract kinetic parameters $k_{on}$ and $k_{off}$, from which the affinity constant $K_D$ of the interaction can be calculated. Virus concentration and time at which dissociation was initiated were constrained to constant values of 16 nM and 24 min, respectively.

**β1 integrin staining on Lec2 cells.** Lec2 cells were stained to visualize β1 integrins using indirect immunofluorescence as described[53]. Confluent Lec2 cells cultivated on glass-bottom dishes were fixed using 4% formaldehyde in PBS for 10 min, washed five times with PBS, and permeabilized using 0.1% Triton X-100 (Sigma-Aldrich) for 1 min, followed by five times washes with PBS. To prevent nonspecific antibody binding, cells were pre-treated with 1% BSA in PBS (Sigma-Aldrich) for 30 min, and then incubated for 1 h with a 1:200 dilution of α5β1 integrin antibody (P1F6, Abcam, ab177004). After another blocking step with 1% BSA for 1 h, cells were incubated with a 1:1000 dilution of Alexa Fluor 647-conjugated goat anti-mouse secondary antibody (Abcam, ab150115) for 1 h. Cells were washed five times with PBS after each blocking step and antibody incubation, and finally washed ten times with PBS before storage at 4 °C until further use. Before imaging, all samples were stained with Hoechst 33342 for 5 min (staining of the cell nucleus at 2 µg mL$^{-1}$ concentration, Thermo Fisher) to facilitate identification of the correct focal plane. After a final wash with PBS, samples were imaged using the 561 nm laser line of an inverted confocal microscope (LSM 980, Zeiss) fitted with a 63× oil objective. Images were acquired through different planes over a 9.845 µm height of cells divided into 16 slices (Z-stack) and covering a 75 × 75 µm area at a resolution of 908 × 908 pixels (optimal resolution based on the scanned area; automatic value set by the software). Integrins (red) and nuclei (blue) were visualized using antibody and DAPI fluorescence, respectively, and contours of cells were visualized using the PMT channel.

**Antibody staining of T3 reovirus particles on cantilever tips.** Individual AFM cantilevers functionalized with virus were placed in the wells of a 24-well plate (Corning) and incubated at room temperature for 1 h with 500 µL blocking buffer (PBS containing 3% BSA). An antibody against serotype 3 reovirus σ1 protein (9BG5, Sigma, MAB994, 0.15 mg mL$^{-1}$)[54] was diluted 1:200 in blocking buffer. Each cantilever was incubated in 500 µL of the primary antibody solution at room temperature for 1 h. Cantilevers were washed three times with blocking buffer. A secondary antibody solution was prepared by diluting a rat anti-mouse IgG2a antibody conjugated to allophycocyanin fluorophore (Thermo Fisher) 1:400 in blocking buffer. Cantilevers were incubated with 500 µL of the secondary antibody solution at room temperature for 1 h. Finally, cantilevers were washed three times in PBS and stored at 4 °C in the dark until further use. The cantilevers were imaged using the 488 nm laser line of an inverted confocal microscope (Zeiss LSM 980).

**Functionalization of gold NPs with reovirus particles.** Gold NPs of 400 nm diameter (Sigma) were functionalized with fluorescently labelled virus (T3SA+ virions or ISVP labelled with Alexa 647 dye) following a three-step protocol.

*COOH functionalization.* A stock solution of alkanethiols was prepared by mixing 600 µL of 1 mM 11-mercapto-1-undecanol solution (Sigma, 2.0 mg dissolved in 10 mL ethanol), 80 µL of 1 mM 16-mercaptohexadecanoic acid (Sigma, 2.8 mg dissolved in 10 mL ethanol) and 120 µL of ethanol. 500 µL of NP suspension (nominal concentration $10^8$ NP mL$^{-1}$) was pelleted by centrifugation (10 min, 12000 rcf), supernatant was removed, the pellet was resuspended in 200 µL of alkanethiols stock solution, and incubated overnight at room temperature under constant shaking.

*NHS-EDC activation.* The functionalized NPs were washed twice by centrifugation (20 min, 12000 rcf) followed by removal of the supernatant and resuspension in 200 µL of ethanol. After the second centrifugation step, NPs were resuspended in 200 µL of an NHS-EDC solution (10 mg mL$^{-1}$ NHS and 25 mg mL$^{-1}$ EDC in milliQ water) and incubated for 30 min under shaking.

*Virus functionalization.* 10 µL of fluorescently labelled virus solution (~$10^{11}$–$10^{12}$ particles mL$^{-1}$ in virion-storage buffer) was added to the NP suspension followed by 60 min incubation under shaking. The NPs were washed by centrifugation (20 min, 12000 rcf) and finally resuspended in 200 µL of fresh virion-storage buffer. The virus-functionalized particles were stored at 4 °C in the dark and used on the same day.

**Functionalization of gold nanoparticles with Alexa 700 dye.** For control experiments, NPs with a diameter of 400 nm (Sigma) were functionalized with an Alexa 700 dye (Invitrogen) following a two-step protocol[29].

*Cysteamine functionalization.* 500 µL of NP suspension ($10^8$ particles mL$^{-1}$) were pelleted by centrifugation (10 min, 12000 rcf) to remove the supernatant, resuspended in 200 µL of a cysteamine solution (Sigma, 100 mM in ethanol), and incubated overnight under shaking.

*Alexa 700 dye functionalization.* The NPs were washed twice by centrifugation (20 min, 12000 rcf) followed by removal of the supernatant and resuspension in

200 µL of ethanol. After the second centrifugation step, the NP were resuspended in 200 µL of NaHCO$_3$ (pH 8.5). 2.5 µL of Alexa Fluor$^{TM}$ 700 NHS Ester (10 mg/mL, Invitrogen, #A20010) was added to the NP suspension, followed by 30 min incubation under shaking in the dark. The NPs were washed by centrifugation (20 min, 12000 rcf) and resuspended in 200 µL NaHCO$_3$, and kept in the dark until further use.

**Assays to probe a functional role for integrin binding in dynamic clathrin recruitment.** A functional role for integrin binding in reovirus internalization was assessed using an instrument combining a FluidFM setup (Cytosurge), a Bruker Resolve AFM, and a Zeiss LSM 980 fast-scanning confocal microscope[29].

To free up integrins adhering to the Petri dish for reovirus binding (i.e., to promote localization to the top surface of the cell and accessible to the FluidFM tip), glass-bottomed Petri dishes used to cultivate cells were coated with Concanavalin-A (ConA) to favour non-specific attachment. Dishes were coated with 2 mL of ConA solution (2 mg/mL) for 2 min and the solution was removed. Dishes were dried in a cell-culture hood for more than 30 min before cells (CHO-JAM-A mTagBFP2-clathrin) were seeded. Experiments were conducted right after seeding to ensure a uniform distribution of the integrins on the cell surface and before integrins migrate toward the bottom of the petri dish to promote cell spreading and attachment.

Virus functionalized NPs were prepared as described above and added to a glass-bottomed Petri dish. They were localized using confocal microscopy and a single NP was trapped into a FluidFM micro-channelled cantilever by applying a steady negative pressure of −800 mbar through the microfluidics system which brings the cantilever in contact with the NP. The NP was then brought in contact with cells, taking advantage of the force control of the AFM. The initial force was set to 500 pN and then decreased to 300 pN and maintained constant for 10 min. While the virus-coated NPs were maintained in contact with the cell surface, time series images were acquired using confocal microscopy. Experiments were conducted using T3SA+ or IBM mutant virions in presence of 2 mM of $Mn^{2+}$. In addition, images were acquired under four conditions: after a 1 h incubation of cells with RGD and KGE blocking peptides (1 mg/mL), after injecting 1 mM Neu5Ac, and using a NP functionalized with T3SA+ ISVP or dye alone.

Confocal images were exported using Zen Blue 3.1 software (Zeiss) and images were analyzed using ImageJ/Fiji (v.1.52e, Time series analyzer). For each time-lapse series of images, a circular ROI was defined around the virus-functionalized NP (ROI NP), and average intensity values from the clathrin fluorescence channel were extracted from each image. To compensate for photobleaching over the 10 min laser exposure, clathrin fluorescence intensity values were extracted in parallel from another circular ROI of the same size defined on a cell area far from the NP contact (ROI background). The ratio of average clathrin fluorescence intensity signals from the two ROIs (ROI NP/ROI background) was finally plotted as a function of time. Intensity vs. time plots were fit to a linear function using Origin 2019 software to extract the slope of the fitted curve, which was used as a comparative parameter to evaluate variations in clathrin recruitment under different experimental conditions.

**Reporting summary.** Further information on research design is available in the Nature Research Reporting Summary linked to this article.

## Data availability
The Source data underlying Figs. 2c, g; 3b–e, g, h; 4b, c, e, f, h; 5b, c, e, f, h; 6c–e and Supplementary Figs. 4e, g; 5 are provided as a Source Data file. The full sequence of pT7-S1T3SA+ is available in Genbank accession no. EF494441. Any other listed plasmids encoding reovirus gene segments or mTagBFP2 can be brought commercially at Addgene. All other relevant data are available from the corresponding authors upon reasonable request.

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

## Acknowledgements

This work was supported by the Université catholique de Louvain, the Fonds National de la Recherche Scientifique (FRS-FNRS) and the Erwin-Schroedinger Fellowship Abroad (FWF Austria, M.K.). S.J.L.P., A.C.D. and D.A. are Research Fellow, Postdoctoral Researcher, and Research Associate at the FNRS, respectively. C.L.G is a European Molecular Biology Organization post-doctoral fellow (EMBO ALTF-2018-542). This project received funding from the European Research Council under the European Union's Horizon 2020 research and innovation program (grant agreement no. 758224) and from the FNRS-Welbio (Grant # CR-2019S-01). Additional support was provided by U.S. Public Health Service awards R01 AI038296 and R01 AI118887 (P.A. and T.S.D.), UPMC Children's Hospital of Pittsburgh (P.A.), and the Heinz Endowments (T.S.D.). The funders had no role in study design, data collection and analysis, decision to publish, or preparation of the paper. The cartoon in Figure 7d was created with BioRender.com.

## Author contributions

M.K., S.J.L.P., J.Y. P.A., T.S.D., and D.A. conceived the project, planned the experiments, and analyzed the data. M.K. and S.J.L.P. conducted the AFM and BLItz experiments. The latter was assisted by M.A.P. A.C.D. supported this study with valuable general input. Fluid-FM and confocal experiments were set up by C.L.G. and carried out by J.Y. P.A. and X.S. engineered and recovered wildtype and mutant viruses. All authors wrote the paper.

## Competing interests

The authors declare no competing interests.
