## [Peer Review File · Nature Communications]

Reviewer comments, first round –

Reviewer #1 (Remarks to the Author):

In this manuscript, the authors focus on a force spectroscopy approach to better understand how reoviruses interact with target cells by interaction with integrins. This is largely based on methods that are well established in the corresponding author's (Alsteen's) lab and they are here carried out according to the state of the art. Their approach is here complemented by measurements in which a Fluid FM approach is used to trap virus-bound nanoparticles and bring them into contact with host cells, which monitoring relevant signalling proteins by confocal microscopy.

Overall the manuscript is technically sound and reads well. I find it hard to comment on its significance, however, as I am not sufficiently familiar with the scientific literature on reovirus entry into the host cells, which will need to be assessed by (an)other reviewer(s).

More specific comments are as follows:

- In Fig. 3c, caption, it would be good to more explicitly specify how "data from single interactions" are objectively distinguished from data resulting from "more than one" interaction.
- I feel it would be beneficial to be a bit more explicit and specific about the "new perspectives" alluded to in the abstract.
- Overall, it would be good to more explicitly recognise that the structural insights here result from rather indirect evidence

Reviewer #2 (Remarks to the Author):

The manuscript of Koehler et al studies the reovirus infectious pathway, or more specifically the cell binding part of this pathway. It is a follow-up of a previous paper (ref 11) in which the binding of reovirus to JAM-A was studied. Now the authors asked themselves what the role of integrin is in binding and uptake. Integrin is best known for its function in focal adhesions, mediating the possibility of cells to attach to surfaces and to crawl over it, but a variety of viruses also use integrin as (co-)receptor for cell binding and/or entry. Especially as it was unclear what the exact role of integrin is for reovirus infection, this was a very relevant question. The authors show with a solid, profound approach based on single-virus AFM force spectroscopy that integrins indeed play an important role. The study is nicely complemented by fluidic force microscopy experiments to study the link between integrin interactions and clathrin recruitment in reovirus infection. While overall the manuscript is clear and appealing, there are certain points to should be addressed.

Fig S1, Here a scratching experiment is performed, but only the "after" image is shown. This image shows blobs at several places and leaves room for various interpretations. Please also include the "before" image (also with height profile)

Line 117 For all percentages and other relevant numbers throughout the manuscript, please use one significant digit for the error, i.e. $19 \pm 7\%$ instead of $18.7 \pm 7.2\%$.

Line 144: The authors conclude that the conformation of integrin on cells alters the binding affinity to reovirus. This conclusion is drawn from experiments performed under differing cation concentrations. The effect of changing ion concentrations on isolated integrins is known, however, if one changes the ion concentration and composition around cells, many other cellular components and processes can change as well. The authors could discuss to what degree other effects than a change in integrin conformation could have led to the observed results.

Line 307-309: Ref 14 does not say that adenovirus undergoes integrin-mediated internalization.

Fig 2b, a scale bar is missing

Fig 3g & h, why are the experiments on model surfaces only performed for low LR and the experiments on cells for high LR? It would be good to have the experiments on model surfaces also performed at high LR

Reviewer #3 (Remarks to the Author):

The paper by Koehler et al makes a highly significant contribution to the understanding of reovirus entry, and more broadly to understanding the complex mechanisms used by nonenveloped viruses during entry.

The set of experiments that is described here moves beyond the functional or observational studies that are viewed as state of the art in viral entry studies, to directly expose mechanism. It is an outstanding example of a study where the scientific question drives the use of advanced technology (AFM), to answer a question of biological relevance. The result is a tour de force, revealing the detailed process whereby reoviruses can make use of different strategies for entry depending on the exact nature of the cell surface environment, and the availability of potential receptor moieties, and in the right circumstances can directly interact with integrin to activate clathrin-mediated internalization of virus. This is a very unusual piece of work in the way that it integrates advanced imaging and viral protein function.

The work rests on a great body of prior work by these investigators, for example the groundwork to understand the effect of Neu5Ac on inducing the conformational change of the attachment protein $\sigma 1$ that mediates initial receptor attachment, which then enhances JAM binding activity. Only on the background of this chain of mechanistic work could the present study have been designed to elucidate the regulation and complexity of interaction with integrin during entry.

The movies are extremely clear and are in no way superfluous, but truly add to the reader's picture of what is happening, and complement the well-designed figures.

Two points merit some additional explanation in the discussion.

1. The dissection of environmental cation effect is clear and rigorous in the results and figures 2,3. The discussion mentions that these results point to physiological relevance of the data, and it would be helpful if this could be spelled out here in the discussion. How do the authors envision cation concentration operating in the physiological setting, with respect to viral entry.

2. The mention of the difference between viruses that engage in high-avidity binding to integrins vs. those viruses that preferentially engage sialic acid is brief in the discussion. It would be helpful to summarize here how the authors envision the interplay or relationship between these different types of virions (the intact virions), and speculate on the place of these differences in nature.

The paper is significant, rigorous, well written, and of broad interest.

Point-by-Point Response to the Reviewers Comments

Reviewer #1 (Remarks to the Author):

In this manuscript, the authors focus on a force spectroscopy approach to better understand how reoviruses interact with target cells by interaction with integrins. This is largely based on methods that are well established in the corresponding author's (Alsteen's) lab and they are here carried out according to the state of the art. Their approach is here complemented by measurements in which a Fluid FM approach is used to trap virus-bound nanoparticles and bring them into contact with host cells, which monitoring relevant signalling proteins by confocal microscopy.

Overall, the manuscript is technically sound and reads well. I find it hard to comment on its significance, however, as I am not sufficiently familiar with the scientific literature on reovirus entry into the host cells, which will need to be assessed by (an)other reviewer(s).

Authors: Thank you for your encouraging and constructive reviews. Below we have explained point-by-point how we have addressed your specific comments.

1) In Fig. 3c, caption, it would be good to more explicitly specify how "data from single interactions" are objectively distinguished from data resulting from "more than one" interaction?

Authors: We thank the reviewer for this suggestion. Generally, to gain full insight into the energy landscape of interactions probed by AFM and determine the multivalency of this interaction, several force distributions are obtained depending on the applied loading rate (LR, force applied over time), leading to the DFS plots shown in Fig. 3b and c. To objectively determine whether single or multiple bond rupture between T3SA+ virions and integrins are taking place, bond strengths (every single grey data point in Fig. 3b and c) were analyzed through distinct discrete ranges of LRs, plotted as force histograms (as shown below in Fig. R1) and further fitted with multi-peak Gaussian distribution, as described^{1,2}. Using these distributions, we are able to determine the most probable unbinding force of each force peak (maximum of rupture force distribution) and put them back in the DFS plot (black dots plotted over mean LR of this range in Fig. 3b and c). The DFS plot shows a linear dependency of the force with the LR, which has been described for other receptor-ligand bonds and is explained by a single free-energy barrier being crossed during the mechanical pulling. With respect to the theory of force spectroscopy, by applying the Bell-Evans model (only valid for single-bond rupture) we can extract the kinetic parameters of single bonds (straight line in Fig. 3b and c, actual fit of the data) and use the predictive Williams-Evans model to determine whether the

multiple bonds are correlated or not, i.e., if all the bonds are equivalent or if the first binding event induces an allosteric modulation of the second binding event. Therefore, if the maxima of the second, third, ... peak of the Gaussian distribution overlay with the Williams-Evans prediction (dotted line in Fig. 3b and c), then we can conclude that multivalent interactions are established as uncorrelated bonds loaded in parallel. We added a more detailed explanation on this matter in the methods section (lines 499-506).

Figure R1: Representative force histogram of one distinct loading rate range

2) I feel it would be beneficial to be a bit more explicit and specific about the "new perspectives" alluded to in the abstract.

Authors: We thank the reviewer for this suggestion. We added another sentence (lines 359 to 361) to the manuscript to be more explicit about the new perspectives hinted at in the abstract. The paragraph now reads as follows:

“Findings reported here shed new light on the complex interplay of interactions between reovirus capsid components and $\beta 1$ integrin in promoting virus entry. A wide variety of viruses, including adenoviruses, coronaviruses, picornaviruses, and reoviruses, have adopted similar mechanisms to enter host cells. A better understanding of these mechanisms will illuminate commonalities underlying pathogen-host interactions that may provide valuable insights for the development of broad-spectrum antiviral therapeutics. In addition, these findings might contribute to the development of more effective virus-based oncolytic treatments by enhancing the targeting of cancer cells.”

3) Overall, it would be good to more explicitly recognise that the structural insights here result from rather indirect evidence.

Authors: We thank the reviewer for this comment and concur that this point should be clarified.

It is thought based on previous studies that reovirus protein $\lambda 2$ is the integrin-binding protein in VPs as well as in ISVPs and cores (probably with different affinities for each particle type). Thus, we first probed the interaction between $\beta 1$ integrins and different viral particles (VPs, ISVPs and cores) and observed differences in binding behavior, which we attribute to the different $\lambda 2$ conformations in VPs, ISVPs, and cores (Fig. 1). We agree that our conclusions and structural insights are made based on indirect evidence. However, we also probed the interaction between integrins and mutants lacking the RGD and KGE integrin-binding motifs in $\lambda 2$, in which we could not detect specific interactions. This finding provides clear evidence that integrin binding to reovirus particles requires intact $\lambda 2$ RGD and KGE motifs. Admittedly, this information is not fully comparable to structural information provided by cryoEM or X-ray crystallography, but we show for the first time in a physiological relevant manner that reovirus binds to $\beta 1$ integrins via the $\lambda 2$ capsid protein. Consequently, we added the following sentence in lines 345 to 346.

“Conclusions drawn from a structural point of view await high-resolution structural analyses for conformation.”

Reviewer #2 (Remarks to the Author):

The manuscript of Koehler et al studies the reovirus infectious pathway, or more specifically the cell binding part of this pathway. It is a follow-up of a previous paper (ref 11) in which the binding of reovirus to JAM-A was studied. Now the authors asked themselves what the role of integrin is in binding and uptake. Integrin is best known for its function in focal adhesions, mediating the possibility of cells to attach to surfaces and to crawl over it, but a variety of viruses also use integrin as (co-)receptor for cell binding and/or entry. Especially as it was unclear what the exact role of integrin is for reovirus infection, this was a very relevant question. The authors show with a solid, profound approach based on single-virus AFM force spectroscopy that integrins indeed play an important role. The study is nicely complemented by fluidic force microscopy experiments to study the link between integrin interactions and clathrin recruitment in reovirus infection.

While overall the manuscript is clear and appealing, there are certain points that should be addressed.

Authors: We thank the reviewer for this constructive, critical, and encouraging appraisal. Below we explain point-by-point how we have addressed the specific concerns raised.

1) Fig S1, Here a scratching experiment is performed, but only the "after" image is shown. This image shows blobs at several places and leaves room for various interpretations. Please also include the "before" image (also with height profile).

Authors: We apologize for this missing information. We have added the image and height profile before scratching.

Supplementary Figure 1 | Validation of surface characteristics and virus immobilization on AFM tip. (a) AFM topography image of an integrin-coated surface before (left) and after (right) scanning a 500 x 500 nm area in the center at a high force (~ 18 nN) to remove attached molecules (referred to as the “scratching” experiment). Insets: Plot of surface thickness along the white dashed line before and after scratching. The extracted profile after scratching shows an accumulation of material along the sides of the scratched square. The biomolecule-free surface inside the square was ~ 3 nm lower than the surrounding biomolecule-coated surface, providing an estimate of the integrin coating thickness. (b) Image of an AFM tip functionalized with reovirus obtained using laser-scanning optical microscopy after staining with a primary antibody against reovirus and an APC-conjugated secondary antibody (red). The inset image shows the virion at the tip apex. Experiments were repeated three times and yielded similar results.

2) Line 117 For all percentages and other relevant numbers throughout the manuscript, please use one significant digit for the error, i.e. $19 \pm 7\%$ instead of $18.7 \pm 7.2\%$.

Authors: We made the requested changes throughout the manuscript, tables, as well as Figs. 2, 3, 4, and 5.

3) Line 144: The authors conclude that the conformation of integrin on cells alters the binding affinity to reovirus. This conclusion is drawn from experiments performed under differing cation concentrations. The effect of changing ion concentrations on isolated integrins is known, however, if one changes the ion concentration and composition around cells, many other cellular components and processes can change as well. The authors could discuss to what degree other effects than a change in integrin conformation could have led to the observed results.

Authors: As pointed out by the reviewers, integrins are a family of α/β heterodimeric adhesion metalloprotein receptors with functions highly dependent on and regulated by different divalent cations. Changes in the extracellular concentration of divalent cations would have many effects on cell and tissue properties influencing their signaling, adhesive, and migratory properties.

Nevertheless, different signaling pathways can initiate integrin activation. In the context of reovirus infection, inside-out signaling could play an important role. For example, intracellular signals received by other receptors can stimulate the binding of adaptor proteins (e.g., talin

and kindlin) to integrin cytoplasmic tails and promote integrin conformational change to a higher affinity state.

4) Line 307-309: Ref 14 does not say that adenovirus undergoes integrin-mediated internalization.

Authors: Thank you for pointing out this mistake, we corrected the sentence as follows and added the correct reference⁵.

“Of particular note is adenovirus, which shares similarities with reovirus in structures of attachment proteins and receptors¹⁴ and also undergoes integrin-mediated internalization³³.”

5) Fig 2b, a scale bar is missing.

Authors: We have added a scale bar.

6) Fig 3g & h, why are the experiments on model surfaces only performed for low LR and the experiments on cells for high LR? It would be good to have the experiments on model surfaces also performed at high LR.

Authors: The difference in the two experimental settings is due to the applied AFM mode suitable for acquiring FD-curves on model surfaces and living cells, respectively. In our experiments using living cells, the data were acquired with the FD-based AFM mode that operates at higher frequency (hence higher LR), enabling us to record more pixels within the map (256-by-256 pixels in this mode compared with 32-by-32 pixels for the mode used for the model surfaces). This approach results in increased lateral resolution of the map and facilitates localization of receptors on the cell. However, we conducted additional experiments on model surfaces at higher LR [fast force volume (FFV) mode] and added an additional 408 data points to Fig. 3b and g (Mn^{2+}) and an additional 227 data points to Fig. 3c and h (Mg^{2+}). These data points are shown as blue dots in the figure below and kept in grey in the manuscript to maintain consistency. The FFV data aligns significantly ($P < 0.0001$) with the model surface and cells and bridges those experiments. The newly extracted values for k_{off} , x_u , and consequently the calculated K_D value did not show a significant change ($P < 0.0001$). However, we updated the values in the manuscript.

(f-h) Assessment of cation-dependent reovirus interaction with integrins expressed on living cells. DFS plots of data obtained using model surfaces in FV mode (grey circles), model surfaces in fast FV mode (blue dots) and living cells (red dots) in the presence of either Mn^{2+} (g) or Mg^{2+} (h). Histogram of the force distribution observed on cells and a multi-peak Gaussian fit of data ($n=900$ data points) are shown at the side. Error bars indicate s.d. of the mean value. All data are representative of $N=5$ independent experiments.

Reviewer #3 (Remarks to the Author):

The paper by Koehler et al makes a highly significant contribution to the understanding of reovirus entry, and more broadly to understanding the complex mechanisms used by nonenveloped viruses during entry.

The set of experiments that is described here moves beyond the functional or observational studies that are viewed as state of the art in viral entry studies, to directly expose mechanism. It is an outstanding example of a study where the scientific question drives the use of advanced technology (AFM), to answer a question of biological relevance. The result is a tour de force, revealing the detailed process whereby reoviruses can make use of different strategies for entry depending on the exact nature of the cell surface environment, and the availability of potential receptor moieties, and in the right circumstances can directly interact with integrin to activate clathrin-mediated internalization of virus. This is a very unusual piece of work in the way that it integrates advanced imaging and viral protein function.

The work rests on a great body of prior work by these investigators, for example the groundwork to understand the effect of Neu5Ac on inducing the conformational change of the attachment protein $\sigma 1$ that mediates initial receptor attachment, which then enhances JAM binding activity. Only on the background of this chain of mechanistic work could the present study have been designed to elucidate the regulation and complexity of interaction with integrin during entry.

The movies are extremely clear and are in no way superfluous, but truly add to the reader's picture of what is happening and complement the well-designed figures.

The paper is significant, rigorous, well written, and of broad interest.

Two points merit some additional explanation in the discussion.

Authors: We thank the reviewer for this positive and encouraging report on our work. Below we explain point-by-point how we have addressed the specific discussion points in our revised manuscript.

1) The dissection of environmental cation effect is clear and rigorous in the results and figures 2,3. The discussion mentions that these results point to physiological relevance of the data, and it would be helpful if this could be spelled out here in the discussion. How do the authors envision cation concentration operating in the physiological setting, with respect to viral entry?

Authors: In the context of this study, we used various cations (Ca^{2+} , Mg^{2+} , Mn^{2+}) to investigate specific interactions between reovirus and integrins and test the physiological relevance of our results. We discovered that in addition to specific binding, the affinity of the interaction is dependent on the activation state of the integrin, which can be influenced by incubation with different cations. However, *in vivo*, other than in particular circumstances (e.g., wound healing), the concentration of divalent cations would not change to a great extent. However, we think that other effects, such as inside-out signaling, could affect the conformation and affinity state of integrins, which could influence viral entry.

2) The mention of the difference between viruses that engage in high-avidity binding to integrins vs. those viruses that preferentially engage sialic acid is brief in the discussion. It would be helpful to summarize here how the authors envision the interplay or relationship between these different types of virions (the intact virions) and speculate on the place of these differences in nature.

Authors: We thank the reviewer for encouraging us to summarize and discuss the results in more detail and also with respect to our previous findings.

In this study, we found that reovirus binding to $\beta 1$ integrins is an important step in virus internalization using different endocytosis pathways via inside-out signaling. We discovered that this process is dependent on (i) the virus particle and (ii) the conformation/activation of the integrin. The internalization scenario dependent on clathrin-mediated endocytosis takes place when virions efficiently bind $\beta 1$ integrins using the RGD/KGE sequences in the $\lambda 2$ capsid protein (through “high-avidity” binding), which in turn induces recruitment of clathrin, promoting virus internalization. We think that caveolin-mediated endocytosis could take place for virions in the presence of abundant free α -SA or ISVPs that bind poorly to integrins (“low-avidity” binding), potentially because of steric hindrance from conformational changes in $\sigma 1$. While it is difficult to speculate about settings in which one pathway will be predominant, capsid proteolysis (leading to ISVPs) will be initiated in the intestinal tract and likely the tumor microenvironment, and integrin binding will be limited, which would favor the caveolin-mediated pathway.

References

- 1 Alsteens, D. *et al.* Nanomechanical mapping of first binding steps of a virus to animal cells. *Nat. Nanotechnol* **12**, 177-183 (2017).
- 2 Delguste, M. *et al.* Multivalent binding of herpesvirus to living cells is tightly regulated during infection. *Sci. Adv.* **4**, eaat1273 (2018).
- 3 Gahmberg, C. G. *et al.* Regulation of integrin activity and signalling. *Biochimica et Biophysica Acta (BBA) - General Subjects* **1790**, 431-444 (2009).
- 4 Tiwari, S., Askari, J. A., Humphries, M. J. & Balleid, N. J. Divalent cations regulate the folding and activation status of integrins during their intracellular trafficking. *J. Cell Sci.* **124**, 1672-1680 (2011).
- 5 Huang, S., Kamata, T., Takada, Y., Ruggeri, Z. M. & Nemerow, G. R. Adenovirus interaction with distinct integrins mediates separate events in cell entry and gene delivery to hematopoietic cells. *J. Virol.* **70**, 4502-4508 (1996).

Reviewer comments, second round –

Reviewer #1 (Remarks to the Author):

In their rebuttal and revised submission, the authors have addressed all points raised in my previous report.

Reviewer #2 (Remarks to the Author):

The authors have addressed all but one of my comments satisfactorily.

It would be good to have a short discussion in the manuscript discussing the following comment:
3) Line 144: The authors conclude that the conformation of integrin on cells alters the binding affinity to reovirus. This conclusion is drawn from experiments performed under differing cation concentrations. The effect of changing ion concentrations on isolated integrins is known, however, if one changes the ion concentration and composition around cells, many other cellular components and processes can change as well. The authors could discuss to what degree other effects than a change in integrin conformation could have led to the observed results.

Manuscript NCOMMS-20-40042-T "Reovirus directly engages integrin to recruit clathrin for entry into host cells" Koehler *et al.*

Point-by-Point Response to the Reviewer's Comment

Reviewer #2 (Remarks to the Author):

The authors have addressed all but one of my comments satisfactorily.

It would be good to have a short discussion in the manuscript discussing the following comment:

3) Line 144: The authors conclude that the conformation of integrin on cells alters the binding affinity to reovirus. This conclusion is drawn from experiments performed under differing cation concentrations. The effect of changing ion concentrations on isolated integrins is known, however, if one changes the ion concentration and composition around cells, many other cellular components and processes can change as well. The authors could discuss to what degree other effects than a change in integrin conformation could have led to the observed results.

Authors: As kindly asked from the reviewer, we included the following short discussion in the manuscript (lines 326-332).

"In addition to altering integrin conformation, it is possible that changes in cation concentration could influence other steps in reovirus entry, e.g., affinity of the virus for other host factors required for internalization and sorting in the endocytic pathway, capsid disassembly, or membrane penetration. However, a significant reduction of cation-mediated effects on reovirus binding and internalization by cRGD would suggest that the effects of cation on reovirus internalization are likely mediated through induced conformation changes in integrin."